



# Combined preliminary-detailed design of wind turbines

Pietro Bortolotti[1], Carlo L. Bottasso[1,2], and Alessandro Croce[2]

[1]Wind Energy Institute, Technische Universität München, D-85748 Garching b. München, Germany
[2]Dipartimento di Scienze e Tecnologie Aerospaziali, Politecnico di Milano, I-20156 Milano, Italy

*Correspondence to:* Carlo L. Bottasso (carlo.bottasso@tum.de)

**Abstract.** This paper is concerned with the holistic optimization of wind turbines. A new procedure is presented that marries for the first time the overall sizing of the machine in terms of rotor diameter and tower height (often termed *preliminary design*), with the detailed sizing of its aerodynamic and structural components. The proposed combined preliminary-detailed approach sizes the overall machine, while taking into full account the subtle and complicated couplings that arise due to the mutual effects of aerodynamic and structural choices. As part of the approach, rotor and tower are sized simultaneously, even in this case capturing the mutual effects of one component over the other due to the tip clearance constraint. Result of the procedure, which is here driven by detailed models of the cost of energy, is a complete aero-structural design of the machine, including its associated control laws.

The proposed methods are tested on the redesign of two wind turbines, a 2.2 MW onshore machine and a large 10 MW offshore one. In both cases, the optimization leads to significant changes with respect to the initial baseline configurations, with noticeable reductions in the cost of energy.

## 1 Introduction

The size of wind turbines has been steadily growing over the last three decades, following a continuous technological trend aiming at better performance and lower costs. Numerous areas of research and development have been involved in this process, such as rotor aerodynamics, rotor and tower structural design and manufacturing, active and passive load reduction techniques, sensing and advanced control strategies, electromechanical conversion, material technology, and many others. Overall, a very significant body of technological improvements has been proposed and developed over the years, the most successful having been slowly but continuously integrated into commercial machines.

In this context, design has the crucial role of evaluating the various technologies and their influence on the final outcome. In fact, as all innovations will come to a cost (in terms of manufacturing, maintenance, availability, etc.), it is only through the holistic view of design that one can judge whether the benefits offered by a new solution offset or not their inevitable drawbacks. To achieve the goal of designing better machines, there is then a need to develop reliable and comprehensive multi-disciplinary design tools. Such tools, invariably based on suitable simulation models, should be able to describe to a sufficient level of fidelity all the relevant physics, and should capture the important couplings among all involved sub-disciplines. In this multi-disciplinary optimization challenge, the most suitable merit figure driving design optimization is often found to be the cost of energy (CoE) (Ning et al., 2013).





Besides being a complex multi-physics problem, a second challenge of wind turbine design is represented by the different operating conditions that a wind turbine encounters throughout its lifetime, a concept currently being translated by standard certification guidelines into the definition of a large number of design load cases (DLCs). This readily excludes the possibility of a monolithic brute-force optimization approach to the design task, and in turn requires more complex algorithmic

structures. Over the years, several research groups have risen to the challenge of addressing this goal by following different approaches. Most of these studies initially focused on the sole blade design problem, as for example in Maalawi et al. (2003); Jureczko and Pawlak (2005); Xudong et al. (2009). Integrated tools appeared later, leading to the development of the commercial packages FOCUS from the Energy Center of The Netherlands (ECN) (Duineveld, 2008) and HAWTOPT from Danmarks Tekniske Universitet (DTU) (Døssing, 2011). In parallel, the multi-disciplinary research code Cp-Max (Code for Performance

Maximization) was successfully developed integrating a high-fidelity aeroelastic simulator together with optimization algorithms, here again evolving from a mostly structural sizing code to a more comprehensive optimization environment (Bottasso et al., 2011, 2015). More recently, other studies followed a multi-level approach to wind turbine design, but with the same focus of achieving a CoE reduction (Maki et al., 2012; Ashuri et al., 2014).

A distinction is often made between *conceptual* (or *preliminary*) and *detailed* design. In the former case, one typically uses

reduced-order models (often in the form of look-up tables, regressions of historical data, analytical low-fidelity models, etc.) in order to identify some macro-parameters of a system, as for example in the present context the rated power, rotor radius, tower height, etc. This initial preliminary design stage is then followed by a detailed design step. In this second phase, one is concerned with the actual optimal sizing of the various aspects of the system, while keeping the macro-parameters fixed. In our context, this means for example finding the optimal aerodynamic shape of the blade, and performing the associated optimal

structural sizing. This two-step process, that clearly can be iterated, works reasonably well in practice, and in fact it is at the basis of classical airplane design methods that are well rooted in the history of aviation (Roskam, 2003; Raymer, 2012).

This distinction is however artificial, and time is ripe for its elimination. In fact, all aspects, disciplines and systems of a wind turbine are so intimately connected that choosing some important parameters based on simplified methods invariably leads to the risk of missing potentially important effects. For example, changing the rotor diameter has dramatic impacts on

the aerodynamics (and hence power performance of the machine), loads (and hence structural sizing, controls, aeroelasticity, sub-systems, etc.), transportation, manufacturing, and other aspects. It is extremely difficult, if not impossible, to accurately account for all these effects without modeling the underlying physical processes. For these reasons, it is important to develop methods that can choose the macroscopic configuration of a machine, taking into full account the effects that these choices imply also at the level of its detailed sizing.

This paper aims at proposing new comprehensive wind turbine design methodologies by including some macro-parameters such as rotor radius and hub height, among others, in the optimization algorithm, while retaining the ability to simultaneously perform a detailed sizing of the machine aerodynamics and structures, together with their associated control laws. To keep the computational effort within the limits of typical industrial practice, where one should be able to deliver a new design configuration in a matter of hours or tens of hours, the code implements a new nested architecture of the optimization algorithm.



This novel implementation of the code represents a marriage, achieved to the authors' knowledge for the fist time, between preliminary and detailed designs, to the benefit of the overall optimization process.

This paper is organized according to the following structure. Section 2 describes the design methodology, with a brief review of the characteristics of the aeroelastic simulation code reported in §2.1, a detailed description of the architecture and
algorithmic flow of the proposed procedures in §2.2, and finally a brief overview of the cost models used for driving the optimization in §2.3. Then, Sect. 3 reports on the application of the new methods to studies of a commercial scale 2.2 MW onshore wind turbine, reported in §3.1, and of a conceptual 10 MW offshore wind turbine, described in §3.2. The paper is closed by Sect. 4, where conclusions are reported and plans for future work are sketched.

## 2   Design methods

### 2.1   Aeroservoelastic simulator

The core of any wind turbine design tool is a simulation model, which must be able to represent with sufficient accuracy the static and dynamic behavior of the machine under all relevant conditions experienced throughout its lifetime. The aeroservoelastic multibody-based code `Cp-Lambda` (Code for Performance, Loads, Aeroelasticity by Multi-Body Dynamic Analysis (Bottasso and Croce, 2006–2016)) is used in this study. The code, originally developed for rotorcraft applications,
is based on Cartesian coordinates and scaled Lagrange multipliers for the enforcement of constraints, while it performs the forward time integration by an implicit non-linearly unconditionally stable energy decaying scheme. `Cp-Lambda` implements a complete library of elements, including non-linear flexible composite-ready beams, rigid bodies, joints, actuators and sensors. The code is tightly coupled with aerodynamic models based on the classical blade-element momentum (BEM) approach, formulated according to the annular stream-tube theory with wake swirl, including tip and hub loss models, as well as un-
steady corrections and dynamic stall. `Cp-Lambda` implements the design guidelines prescribed by international certification standards (IEC61400-1, 2005; GL, 2010).

The code has been used in several industrial and research projects, and it has been validated against industrial simulation programs, wind tunnel experimental results and field measurements. Readers interested in the mathematical formulation of `Cp-Lambda` can refer to Bottasso et al. (2006); Bauchau et al. (2003, 2009); Bauchau (2011), while wind turbine applications
of the code can be found in Bottasso et al. (2011, 2015), among others.

### 2.2   Wind turbine design algorithm

`Cp-Max` is a wind turbine design tool wrapped around `Cp-Lambda`, and its latest architecture is presented in the following. The code was first implemented as an aerodynamic optimization tool for blade chord and twist distributions aiming at a maximization of the annual energy production (AEP) for a given wind turbine macro configuration. The procedures soon also
included a purely structural optimization package for the blade (Bottasso et al., 2011), whose merit figure was the minimization of rotor mass. This was achieved by coupling `Cp-Lambda` with the finite element cross sectional analysis code ANBA





(ANisotropic Beam Analysis), implementing the theory of Giavotto et al. (1983). Given airfoils, blade topology, composite mechanical properties and the geometry of the cross section structural members, ANBA produces the six-by-six stiffness matrix that defines the sectional characteristics at a given spanwise location of the geometrically-exact shear and torsion-deformable beam model used in Cp-Lambda. A similar procedure allows also for the structural sizing of the wind turbine tower, which

can optionally be dimensioned simultaneously to the rotor (Bottasso et al., 2014a).

Because of the very definition of a beam and a beam cross section, none of these models is capable of capturing three-dimensional effects in regions of very rapid changes or discontinuity in the structural geometry and/or material properties, as for example at stations where shear webs begin or end. To address this intrinsic limitation of combined sectional/beam models, the code was equipped with a multi-level approach, whereby a detailed FEM model of the blade is used to capture

the three-dimensional state of stress and strain to a higher level of precision. Iterations between the sectional-aeroservoelastic and FEM levels are used to ensure that all desired structural constraints (as the satisfaction of allowables, fatigue, buckling, etc.) are verified at the fine FEM level by means of static, modal and fatigue analyses. As more fully described in Bottasso et al. (2014a), the FEM-level analyses are conducted by using loads computed at the aeroservoelastic level, and results of such analyses are used for updating the bounds of design constraints at the next iteration.

A further expansion of the wind turbine design methodology was finally reached when ad-hoc algorithms were formulated to simultaneously optimize blade aerodynamics and structure. This offered the opportunity to perform a truly integrated aero-structural rotor design optimization (Bottasso et al., 2015).

However, this version of Cp-Max represented a detailed design tool that lacked the ability to directly modify the macroscopic configurational parameters of the wind turbine. Extensive use of the software highlighted a general weak sensitivity of the CoE

merit figure to the blade aerodynamic and structural design parameters at frozen global wind turbine configuration, i.e. at fixed rotor diameter and tower height. In other words, while changes in the details of the blade aerodynamic shape and structural components significantly affect AEP, mass, loads etc., in reality CoE often appeared to be significantly flat around an optimum.

The present paper aims at developing procedures for a more extensive exploration of the design space, through a global redesign activity of the wind turbine. This is achieved by including in the optimization process macro parameters that are

typically associated with a preliminary design phase. However, differently from simpler approaches, the inclusion of macro parameters in the optimization is done here while retaining the ability to perform a multi-level aero-structural design of the rotor (and optionally of the tower), achieving in this way the marriage between the preliminary and detailed design phases. This is done with the goal of capturing at the level of the macro design parameters also the effects of the detailed design features, avoiding simplifications and the danger of missing important couplings.

The overall architecture of the resulting multi-level combined preliminary-detailed design procedures, as more precisely described later on in the following pages and in Bottasso et al. (2011, 2014a, 2015), is shown in Fig. 1.

The rest of this discussion is organized as follows. First, the aerodynamic optimization algorithm is briefly presented in §2.2.1. Then, a short summary of the structural optimization is reported in §2.2.2. The proposed nested structure of the combined preliminary-detailed algorithm is finally presented in §2.2.3. In addition, Chaviaropoulos and Sieros (2014) high-



lighted the potential benefits of low induction rotors (LIR). A way to accommodate the design of such rotors in the current framework is discussed in §2.2.4.

In the following sections, for clarity of the formulation, a formal description of the structure of the algorithms is given. To this end, functions are indicated with the notation

$$(O) = \texttt{FunctionName}(I) \tag{1}$$

where $I$ are the input variables, while $O$ the output ones.

### 2.2.1 Aerodynamic optimization

The aerodynamic optimization function, described in detail in Bottasso et al. (2015), is here only rapidly recalled with the following formal description:

**Function** $(\boldsymbol{p}_a^*, AEP^*) = \texttt{MaxAEP}(\boldsymbol{p}_a, \boldsymbol{p}_s, \boldsymbol{p}_g, D):$ $\tag{2a}$

$$\boldsymbol{p}_a = \boldsymbol{p}_{a_c} \cup \boldsymbol{p}_{a_\theta} \cup \boldsymbol{p}_{a_t}, \tag{2b}$$

$$AEP^* = \max_{\boldsymbol{p}_a}\big(\texttt{ComputeAEP}(\boldsymbol{p}_a, \boldsymbol{p}_s, \boldsymbol{p}_g, D)\big)$$

$$\big(\text{and } \boldsymbol{p}_a^* = \arg\big(\max_{\boldsymbol{p}_a}(\texttt{ComputeAEP})\big)\big), \tag{2c}$$

$$\text{s.t.:} \quad \boldsymbol{g}_a(\boldsymbol{p}_a) \leq \boldsymbol{0}. \tag{2d}$$

where $\boldsymbol{p}_a$, $\boldsymbol{p}_s$ and $\boldsymbol{p}_g$ are vector arrays containing, respectively, the aerodynamic, structural and global variables of the optimization problem. Function $\texttt{MaxAEP}$ optimizes $\boldsymbol{p}_a$, while $\boldsymbol{p}_s$ and $\boldsymbol{p}_g$ are controlled by function (4) in §2.2.2 and by function (5) in §2.2.3, respectively. As shown in (2b), $\boldsymbol{p}_a$ includes the three vectors $\boldsymbol{p}_{a_c}$, $\boldsymbol{p}_{a_\theta}$ and $\boldsymbol{p}_{a_t}$ containing discrete nodal parameters that control chord, twist and thickness distributions, respectively, obtained by spline interpolation. The thickness distribution described by $\boldsymbol{p}_{a_t}$ is obtained by interpolating the thicknesses of a given number of chosen airfoils; by controlling their spanwise position, one in turn may affect the local thickness of the blade. Finally, $D$ is a list of given input data:

$$D = \{P_r, C, V_{\text{in}}, V_{\text{out}}, AF, C, v_{\text{tip}_{\max}}, L_{\text{DLC}}, \ldots\}. \tag{3}$$

The list includes all the quantities that remain constant through the different optimization loops such as, among others, generator rated power $P_r$, wind turbine class $C$, cut-in $V_{\text{in}}$ and cut-out $V_{\text{out}}$ wind speeds, blade airfoil family $AF$, maximum allowable tip speed $v_{\text{tip}_{\max}}$ and the list $L_{\text{DLC}} = \{\ldots, \text{DLC}i.j, \ldots\}$ containing all the DLCs (IEC61400-1, 2005; GL, 2010) that one decides to consider in the optimization of the machine.



Goal of the aerodynamic optimization is to achieve the highest annual energy production, whose optimum value is noted $AEP^*$ in (2c), while respecting the non-linear constraints $\boldsymbol{g}_a(\boldsymbol{p}_a)$ expressed by (2d). The vector of conditions $\boldsymbol{g}_a$ can be tailored based on design needs, and it typically includes limits on the maximum allowable tip speed, maximum chord, upper and lower bounds on solidity and tapering for chord and thickness distributions, as well as limitations to the twist distribution in

order to take into account manufacturing constraints. The solution to the constrained optimization problem, which is typically solved by means of a sequential quadratic programming (SQP) optimization algorithm, is indicated as $\boldsymbol{p}_a^*$ in correspondence to the optimum cost $AEP^*$, as shown in (2c).

### 2.2.2 Structural optimization

The structural optimization procedure, described in detail in Bottasso et al. (2015), is a more complex and computationally

expensive loop that aims at identifying the set of parameters $\boldsymbol{p}_s^*$, which describe blade and tower structure at frozen rotor shape, associated with the minimum initial capital cost $ICC^*$. $\boldsymbol{p}_s$ is a vector containing the thickness of the structural components at selected stations along the blades, such as spar caps, skin, shear webs and reinforcements; for the tower, this vector contains the outer diameters and wall thicknesses at selected stations along its height. The corresponding distributions at generic stations are obtained by spline interpolations of these nodal values. The formal description of the algorithm is as follows:

**Function** $(\boldsymbol{p}_s^*, ICC^*) = \texttt{MinICC}(\boldsymbol{p}_a, \boldsymbol{p}_s, \boldsymbol{p}_g, D, \boldsymbol{\Gamma}_s):$         (4a)

   **do**         (4b)

     $(\boldsymbol{E}) = \texttt{LoadEnvelope}(\boldsymbol{p}_a, \boldsymbol{p}_s, \boldsymbol{p}_g, D),$         (4c)

     $ICC^* = \min_{\boldsymbol{p}_s} \big(\texttt{ComputeICC}(\boldsymbol{p}_a, \boldsymbol{p}_s, \boldsymbol{p}_g, D, \boldsymbol{E}, \boldsymbol{\Gamma}_s)\big),$

     $\big(\text{and } \boldsymbol{p}_s^* = \arg\big(\min_{\boldsymbol{p}_s}(\texttt{ComputeICC})\big)\big),$         (4d)

$(\boldsymbol{\Gamma}_s^*) = \texttt{3DFEAnalysis}(\boldsymbol{p}_a, \boldsymbol{p}_s^*, \boldsymbol{p}_g, D, \boldsymbol{E}, \boldsymbol{\Gamma}_s),$         (4e)

     $\Delta p_s = \|\boldsymbol{p}_s^* - \boldsymbol{p}_s\|, \quad \Delta ICC = \|ICC^* - ICC\|, \quad \Delta\Gamma_s = \|\boldsymbol{\Gamma}_s^* - \boldsymbol{\Gamma}_s\|,$         (4f)

     $\boldsymbol{p}_s = \boldsymbol{p}_s^*, \quad \boldsymbol{\Gamma}_s = \boldsymbol{\Gamma}_s^*$         (4g)

   **while** $(\Delta p_s \geq \text{tol}_{p_s}, \ \Delta ICC \geq \text{tol}_{ICC}, \ \Delta\Gamma_s \geq \text{tol}_{\Gamma_s}).$         (4h)

The structural optimization is an iterative loop, which begins with a load computation step, expressed by (4c): here control laws are first computed based on the wind turbine design, and then DLCs from the list $L_{\text{DLC}}$ in (3) are run by using the simulation model (in the present case, implemented in `Cp-Lambda`).

The post-processed results of these analyses are used to compute the load envelopes $\boldsymbol{E}$ at a number of verification stations along blades and tower. The load envelope at each station includes, for each internal stress resultant component, the maxi-



mum positive and minimum negative values encountered within the machine lifetime, and their generating DLC. The rainflow counting required to estimate fatigue damage is also performed here.

This step is followed by a rotor and tower structural sizing for the given load envelopes $\boldsymbol{E}$, as expressed by (4d). In this second step, the minimum initial capital cost $ICC^*$ is computed, together with its associated optimal set of design variables $\boldsymbol{p}_s^*$. The inputs to `ComputeICC` are the aerodynamic, structural and global parameters $\boldsymbol{p}_a$, $\boldsymbol{p}_s$ and $\boldsymbol{p}_g$, respectively, the input list $D$, the load envelopes $\boldsymbol{E}$ at the verification stations, and finally a list of parameters $\boldsymbol{\Gamma}_s$ used to impose desired design requirements. $\boldsymbol{\Gamma}_s$ includes the admissible values for stress and strain, frequency constraints, buckling constraints for sandwich core sizing and the maximum allowable blade tip deflection based on tower clearance (updated based on the current geometry of the machine and the tower). As for the maximum of `ComputeAEP`, the minimum of `ComputeICC` is also solved by means of a SQP optimization algorithm, which is well suited to problems with several constraints that are potentially simultaneously active at convergence.

The structural sizing of `ComputeICC` is followed by (4e), which represents a higher-fidelity 3D FEM analysis, whose role is to verify the fulfillment of all the structural constraints at a finer description level, by updating when necessary vector $\boldsymbol{\Gamma}_s$ into $\boldsymbol{\Gamma}_s^*$ (Bottasso et al., 2014a). Given inner and outer blade geometry, a 3D shell-element mesh of the blade is created and associated with a set of load conditions. Such conditions are obtained by post-processing the outputs of the aeroservoelastic simulation of all considered DLCs, selecting those loads that induce extreme stress and strain values, loads associated with maximum tip deflections, as well as time histories of the turbulent load cases for the evaluation of fatigue damage. For each loading condition, span-wise distributions of the internal stress resultants and of the aerodynamic forces are readily available from the multibody simulations. These are used for computing equivalent loads that are then applied to the FEM model to achieve realistic loading conditions for each blade component, e.g. by limiting the application of the aerodynamic loads to the external skin nodes. The FE input model is then fed to the commercial FE solver NASTRAN (MSC Software, 2012), which is in turn coupled to an automated post processing routine that closes the loop.

Function `3DFEAnalaysis` is generally found to produce changes in the constraint bounds $\boldsymbol{\Gamma}_s$ for the blade root design, the detailed sandwich core sizing and in the presence of large 3D effects, such as blade regions with strong transitions in chord size or at the beginning and end of the shear webs. On the other hand, most of the other blade components are generally well sized by the analysis performed at the beam and sectional levels in (4d). In this sense, $ICC$ is often not largely affected by (4e).

Overall, the structural loop of Eq. (4) converges when $\boldsymbol{p}_s$, $ICC$ and $\boldsymbol{\Gamma}_s$ are within a predefined tolerance, as reported in (4h).





### 2.2.3 Aero-structural optimization

The aero-structural optimization is an outer optimization loop stacking in sequence the aerodynamic optimization, the structural optimization and the CoE evaluation. Its goal is to find the optimal vector $\boldsymbol{p}_g^*$, and the associated aerodynamic and structural vectors $\boldsymbol{p}_a^*$ and $\boldsymbol{p}_s^*$, that achieve a minimum cost of energy $CoE^*$. The algorithm can be formally described as:

**Function** $(\boldsymbol{p}_a^*, \boldsymbol{p}_s^*, \boldsymbol{p}_g^*, CoE^*) = \texttt{MinCoE}(\boldsymbol{p}_a, \boldsymbol{p}_s, \boldsymbol{p}_g, D, \boldsymbol{\Gamma}_s) :$  (5a)

$$CoE^* = \min_{\boldsymbol{p}_g} \big( \texttt{ComputeCoE}(\boldsymbol{p}_a, \boldsymbol{p}_s, \boldsymbol{p}_g, D, \boldsymbol{\Gamma}_s) \big),$$

$$\big(\text{and } (\boldsymbol{p}_a^*, \boldsymbol{p}_s^*, \boldsymbol{p}_g^*) = \arg \big( \min_{\boldsymbol{p}_g}(\texttt{ComputeCoE}) \big) \big),$$  (5b)

$$\text{s.t.:} \quad \boldsymbol{g}_g(\boldsymbol{p}_g) \leq \boldsymbol{0}.$$  (5c)

The vector of global optimization variables $\boldsymbol{p}_g$ is defined as:

$$\boldsymbol{p}_g = [R, H, \gamma, \phi, \sigma_c, \tau_c, \sigma_t, \tau_t],$$  (6)

where the symbols indicate the rotor radius $R$, hub height $H$, rotor cone angle $\gamma$, nacelle uptilt angle $\phi$ and four blade aero-structural terms $\sigma_c$, $\tau_c$, $\sigma_t$ and $\tau_t$. The rotor radius $R$ directly influences the length of the blades, causing cascade changes in the aerodynamic performance of the machine, in its regulation trajectory as well as in the loads. Moreover, $R$ is a scaling

factor for cost items within the CoE models, as for instance the pitch system costs. As a result, the CoE merit figure has the highest sensitivity with respect to $R$.

The tower height $H$ offers the possibility to perform a site-specific wind turbine optimization. One the one hand, a higher $H$ is likely to cause higher tower costs due to a longer tower structure and due to higher fatigue loads. On the other hand, aerodynamic benefits are implied by the wind shear power law, which causes higher average wind speeds and lower shear

at higher $H$. A power coefficient equal to 0.2 is assumed as prescribed by the IEC standards (IEC61400-1, 2005). Given the statistical nature of storm winds, no dependency on $H$ is assumed for the 50-year storm wind speed.

Parameters $\gamma$ and $\phi$ affect both aerodynamics and structure. The power coefficient $C_p$ in fact typically decreases with increasing $\gamma$ and $\phi$, causing a reduction in AEP, while the maximum allowable tip deflection constraint is relaxed at growing $\gamma$ and $\phi$, leading to potential structural benefits.





Finally, the two $\sigma$ parameters are defined as rotor planar solidity $\sigma_c$ and blade thickness solidity $\sigma_t$, while the two $\tau$ parameters are defined as blade planar tapering $\tau_c$ and blade thickness tapering $\tau_t$. Their mathematical expressions are given as follows:

$$\sigma_c = \frac{3A_b}{A} = \frac{3\int_0^R c(r)\,\mathrm{d}r}{\pi R^2}, \tag{7a}$$

$$\tau_c = \frac{\int_0^R rc(r)\,\mathrm{d}r}{A_b}, \tag{7b}$$

$$\sigma_t = \frac{1}{100}\int_0^1 t(\eta)\,\mathrm{d}\eta, \tag{7c}$$

$$\tau_t = \frac{\int_0^1 \eta t(\eta)\,\mathrm{d}\eta}{\int_0^1 t(\eta)\,\mathrm{d}\eta}, \tag{7d}$$

where $A_b$ is the blade planar area, $A$ is the rotor swept area, $c$ the chord, $r$ is the dimensional blade span, $t$ is the blade percentage thickness and $\eta$ the non-dimensional blade span. The role of the four parameters $\sigma_c$, $\tau_c$, $\sigma_t$ and $\tau_t$ is to allow for an interaction between the aerodynamic loop of function (2) and the structural loop of function (4), in turn enabling an integrated rotor aero-structural design optimization. From a computational point of view, they enter as non-linear constraints into the aerodynamic blade shape definition expressed by (2d).

Goal of the integrated optimization is to find the minimum cost $CoE^*$ in (5b), whose computing function `ComputeCoE` can be expressed as:

$$\textbf{Function } (\boldsymbol{p}_a^*, \boldsymbol{p}_s^*, \boldsymbol{p}_g, CoE) = \texttt{ComputeCoE}(\boldsymbol{p}_a, \boldsymbol{p}_s, \boldsymbol{p}_g, D, \boldsymbol{\Gamma}_s): \tag{8a}$$

$$(\boldsymbol{p}_a^*, AEP^*) = \texttt{MaxAEP}(\boldsymbol{p}_a, \boldsymbol{p}_s, \boldsymbol{p}_g, D), \tag{8b}$$

$$(\boldsymbol{p}_s^*, ICC^*) = \texttt{MinICC}(\boldsymbol{p}_a^*, \boldsymbol{p}_s, \boldsymbol{p}_g, D, \boldsymbol{\Gamma}_s), \tag{8c}$$

$$(AEP^{**}) = \texttt{ComputeAEP}(\boldsymbol{p}_a^*, \boldsymbol{p}_s^*, \boldsymbol{p}_g, D)), \tag{8d}$$

$$(CoE) = \texttt{CoEmod}(AEP^{**}, ICC^*, \boldsymbol{p}_a^*, \boldsymbol{p}_s^*, \boldsymbol{p}_g, D). \tag{8e}$$

The procedure is obtained by conducting in sequence an aerodynamic optimization, given in (8b), a structural optimization, given in (8c), a new calculation of the AEP considering the updated structure $\boldsymbol{p}_s^*$, given in (8d), and a final evaluation of the CoE from the cost models, given in (8e) and later discussed in §2.3. The optimization loop also considers optional constraints on load envelopes. This may be necessary in the presence of frozen components, for instance the shaft or the nacelle, for which one may have to express maximum and un-exceedable loads. These constraints are to be defined in (5c).

The architecture of the preliminary-detailed design optimization procedure is shown in Fig. 2. As for the previous sub-problems, even this coupled aero-structural optimization problem is solved using a SQP algorithm.





To limit computational cost, the most expensive operations are parallelized. In particular, DLCs are run in parallel independently on all available cores. The same is done for the gradients in the structural loop of (4d). As no interdependency among these tasks exists, this amounts to a classical case of embarrassing parallelism, which is simply implemented by dispatching jobs on all available computational cores, and the remaining ones on the cores that become available after having completed

their assigned job. It should be noticed that this current simple implementation is not capable of guaranteeing a full utilization of the available resources, as different DLCs have often different physical time lengths (e.g., 600 s for turbulent simulations, a few tens of seconds for gusts, etc.) and hence they typically imply different simulation times. In addition, using a number of cores that is not a submultiple of the number of DLCs leads to further inefficiencies. A more optimized use of the available computational resources is certainly possible, and it should be the focus of future work. As the number of design variables is

relatively small, the solution of the optimization problem is of negligible cost (once constraints and cost function have been evaluated), and therefore it is not parallelized in the current implementation. Depending on the number of DLCs, the number of design variables, and the mesh refinement of the multibody model, the overall design process can be typically completed in a matter of hours or tens of hours.

### 2.2.4 Low induction rotor configuration

The development of multi-disciplinary tools offers the opportunity to explore alternative wind turbine designs. LIRs are one such possible solution, where the wind turbine operates on purpose at a sub-optimal aerodynamic efficiency, potentially benefiting from reduced loads and consequently lighter and cheaper structures.

From an algorithmic point of view, a LIR can be designed within the current framework by means of an offset $\delta$ applied to the pitch angle, so as to feather the blade towards lower angles of attack. Parameter $\delta$ affects both the aerodynamics and the

20 structure of the wind turbine, and therefore it is included in the $\boldsymbol{p}_g$ vector of design variables:

$$\boldsymbol{p}_g = [R, H, \gamma, \phi, \sigma_c, \tau_c, \sigma_t, \tau_t, \delta]. \tag{9}$$

The use of $\delta$ results in a perturbed regulation trajectory with a lower maximum power coefficient $C_p$ in the partial load region. This also implies lower lift and drag aerodynamic forces for wind speeds up to the rated wind velocity. The design challenge is to identify the potential optimum trade-off between losses in aerodynamic efficiency and structural advantages in terms of

25 ICC. The CoE is once again the merit figure to be monitored during this optimization.

### 2.3 Cost of energy models

The ultimate figure of merit for a wind turbine multi-disciplinary optimization process is the CoE. It is therefore clear that accurate CoE models are of crucial importance. In fact, as the CoE drives the design, any inaccuracy in the cost model will invariably affect the design itself. In this work we have made use of the NREL cost model (Fingersh et al., 2006) and the

30 more recent INNWIND one (Chaviaropoulos et al., 2014). The main difference between the two models is the applicability range, as the NREL CoE model was initially developed for mid-size onshore wind turbines and only later adapted to offshore



applications, while the INNWIND CoE model has been especially formulated for multi-MW next-generation offshore wind turbines.

In addition to the two CoE models, a highly detailed blade cost model (BCM) developed at SANDIA National Laboratories by Johans and Griffith (2013) is also implemented in the code. This model is capable of capturing the aero-structural trade-offs

of the rotor and overcomes the simplified relationships between blade mass or blade length versus blade cost used in the NREL and INNWIND CoE models. The SANDIA BCM is in fact composed of three main items: material costs, labor costs and equipment costs. Material costs are estimated based on the mass of each blade structural component, differentiating between the costs of different fibers, resins, sandwich core and extra materials as adhesive, paint, lightning protection, etc. Labor costs estimate the man hours needed for the manufacturing of a single blade, which are then multiplied by the wage rate, a value that

can be readily tuned based for instance on the country of manufacturing. Labor hours are estimated based on reference models and several geometrical and structural scaling factors. Finally, equipment costs are estimated as price of mold and tooling divided by the number of blades that can be manufactured with a single set of equipment. The price of mold and tooling is upscaled using a power law expressed as a function of rotor radius.

## 3    Applications

The combined preliminary-detailed optimization methodology described in Sect. 2 is applied to two reference wind turbine models: a 2.2 MW wind turbine representative of current mid-size commercial-scale onshore machines, and a 10 MW wind turbine representative of large next-generation offshore machines. The design optimization of the 2.2 MW reference machine is presented in §3.1, while the 10 MW wind turbine, originally developed by DTU and released in the public domain for research purposes (Bak et al., 2013) is discussed in §3.2.

### 20    3.1    2.2 MW onshore wind turbine

The 2.2 MW baseline machine is a class 3A onshore three-bladed wind turbine with a steel tower and a standard glass fiber reinforced plastic (GFRP) blade configuration, featuring two spar caps, two shear webs, a skin layer and extra uni-directional (UD) reinforcements at the leading and trailing edges. The main parameters of the wind turbine are reported in Table 1.

Regarding aerodynamics, the blades are equipped with DU airfoils (Timmer and van Rooij, 2003) located as listed in Ta-

ble 2, while the chord and twist distributions are shown in Fig. 3. The structural design, the blade topology and its structural configuration are described in Table 3, while the material mechanical properties are listed in Table 4.

A reduced set of DLCs is selected in order to conduct the optimization design studies. Among the full set of design conditions, DLCs 1.1, 1.2, 1.3, 2.1, 2.3 and 6.2 (IEC61400-1, 2005) were identified as those producing design drivers for the baseline wind turbine, and were therefore assumed to be sufficiently representative of the different load scenarios encountered by the

machine. These DLCs represent normal operating conditions, extreme turbulent wind conditions, the occurrence of extreme gusts combined with electric faults and, finally, the occurrence of a 50-year storm at different yaw angle values. One single wind seed was assumed for the turbulent cases, although four or six seeds are typically considered necessary for certification.





The resulting total number of DLCs used for optimization is 64, which is a relatively small number chosen to contain the computational cost.

While for better accuracy the fatigue DLCs should be increased with respect to the ones used here, it is not always strictly necessary to include a very large number of DLCs in the optimization loop. In fact, we typically run a full set of DLCs
(generally including several hundreds of different conditions) on the resulting final design. In case some of the driving loads change with respect to the ones obtained within the design loop, then we update the list of DLCs $L_{\mathrm{DLC}}$ with the new design-driving ones and we repeat the design. However, this seldom happens if one has made a careful choice at the beginning, unless the design experiences large changes with respect to the starting point.

A load analysis of the baseline configuration returns active constraints for the blade tip deflection during operation, result-
10 ing in a flapwise stiffness-driven blade design, active fatigue constraints for the blade shell skin, as well as active buckling constraints for the steel tower due to storm loads. Frequency constraints for blade and tower are also close to be active.

### 3.1.1 Holistic optimization

The baseline design of the 2.2 MW wind turbine is used as starting point for a full design optimization where the merit figure is the CoE calculated from the NREL cost model, while the blade cost is calculated from the SANDIA BCM. Table 5 reports
the initial and final values of the design parameters $\boldsymbol{p}_g$.

The global trend of the design optimization is a clear upscale of the machine. Thanks to a larger rotor diameter and a taller hub height, a higher energy capture is indeed obtained, leading to significant advantages in terms of CoE. Cone and uptilt angles are also increased to relax the tower clearance constraint and cause the simultaneous activation of both the tip deflection and blade frequency constraints. Finally, the four blade aero-structural parameters are adjusted with respect to their baseline
values to achieve an aero-structural trade-off. The rotor aerodynamic performance is indeed slightly decreased due to the aerodynamically suboptimal chord and thickness distributions shown in Fig. 4; however this allows for a limiting of the ICC caused by the longer blades. Minor modifications are also produced to the twist distribution on account of the different airfoil positioning.

From a blade structural point of view, thicker structural elements are designed to withstand the higher loads. The distributions
for spar caps, skin, webs and trailing edge reinforcement are reported in Fig. 5. Core thickness also exhibits a growth due to larger sandwich panels and higher loads. The resulting blade mass suffers a 51.6% increase. Finally, the coupled optimization of rotor and tower identifies an optimal distribution for the tower diameters in order to balance tower clearance and stiffness, to the benefit of ICC. The distributions of outer diameters and wall thicknesses along the tower height are shown in Fig. 6. The higher and thicker tower is heavier than the baseline by 38.7%.

Overall, the optimization process leads to an increase of 16.5% in the ICC, caused by the growth of rotor, tower, drive-train and nacelle costs, equal to 35.1%, 38.7% and 10.3%, respectively. The higher costs are nevertheless largely compensated by an increase of 20.0% in the AEP, resulting in net savings in terms of CoE of 3.1%. It should be remarked that it would be difficult, if not impossible, to exactly quantify the effects on rotor and tower (which largely depend on their detailed sizing, accounting for all design-driving conditions) caused by changes in the macro parameters (rotor diameter and tower height),



without actually performing a detailed design. Therefore, with a classical approach based on a preliminary design of the macro parameters followed by a detailed design at fixed rotor diameter and tower height, it might have been harder to identify the CoE-optimal solution found here in one single shot.

The final design was obtained after only 4 iterations of the SQP algorithm, with a total computational time of approximately

65 hours on a workstation equipped with 40 logical processors running the Windows 10 operating system. Given that the analysis included 64 DLCs of different physical time lengths, the simulation time can be considered far from optimized, and significant reductions are certainly possible.

### 3.1.2   Low induction configuration

A second study is conducted on the 2.2 MW onshore machine, introducing the pitch offset $\delta$ within the vector of design

parameters $\boldsymbol{p}_g$ (cf. Eq. (9)). This additional degree of freedom allows for the algorithm to choose a LI configuration (operating at lower rotor efficiency), if such a solution turns out to be further improving the merit function with respect to a optimal induction one. Therefore, it is important to remark that this way of approaching the problem does not force a LI solution, which will only appear if and only if it improves the CoE with respect to a non-LI one.

The outcome of this problem setup results in a solution that is identical to the one of the problem exposed in §3.1.1, with an

optimal $\delta$ value equal to 0. This means that there is apparently no advantage in reducing the aerodynamic efficiency to benefit from reduced loads. In fact, savings in the latter are very limited compared to losses in the former, and the small reductions in ICC do not justify drops in the power coefficient. It is therefore concluded that a LI configuration does not improve the design of this specific 2.2 MW wind turbine.

### 3.2   10 MW offshore wind turbine

The proposed methodology is then exercised on the optimization of a large scale wind turbine, representative of the next generation offshore machines. A 10 MW machine, developed in Bottasso et al. (2015) as an evolution of the original DTU 10 MW wind turbine (Bak et al., 2013) is chosen as a significant test case. The main characteristics of the wind turbine are reported in Table 6.

The reference chord and twist distributions are shown in Fig. 7, while the blades are equipped with FFA airfoils (Björck,

1990) positioned as listed in Table 7. The blade topology and the structural configuration are detailed in Table 8, while material properties are summarized in Table 9. The blade has a two spar caps – three webs topology, with UD composite reinforcements at the leading edge, trailing edge and in the root region. Different GFRP laminates are used in the various structural elements, while balsa wood is used as core material in the sandwich panels.

The same set of DLCs used for the 2.2 MW wind turbine is adopted also in the design studies of the 10 MW machine (IN-

NWIND.EU, 2015), while wind conditions are adjusted for its different class following IEC certification guidelines (IEC61400-1, 2005). Namely, the average wind speed at a hub height of 119 m is assumed to be 10 m/s, while the 50-year storm wind speed is set to 50 m/s. The 10 MW reference rotor is found to be highly tip-deflection-driven, with the blade flap frequency constraint largely satisfied. Moreover, the ratio of edge to flap blade frequencies, imposed to be higher than 1.1, drives the edgewise



stiffness, and in turn the design of trailing and leading edge reinforcements. Finally, blade skin is again fatigue driven, while tower structure is designed against buckling caused by storm loads.

### 3.2.1 Holistic optimization

The reference design of the 10 MW offshore wind turbine is used as initial starting guess for a combined preliminary-detailed
optimization study. The merit figure is the CoE computed from a combination of the INNWIND and SANDIA cost models. Overall, the proposed holistic approach finds the reference configuration largely under-designed and performs a significant upscaling of the rotor diameter and hub height. The final rotor design has both the blade frequency and maximum allowable tip deflection constraints that are active at convergence. This results from the combined detailed sizing of rotor and tower, together with the adjustment of rotor cone angle and nacelle uptilt.

A comparison of the elements of the vector of design parameters $\boldsymbol{p}_g$ is shown in Table 10.

    In terms of the blade aero-structural parameters, only the chord distribution is adjusted towards a lower rotor solidity, as shown in Fig. 8, while the airfoil positions remain essentially the same. A check is performed running a new optimization from a perturbed initial guess and very similar results are obtained in terms of blade thickness distribution. The twist also undergoes changes, particularly in the tip region, which in the end cause small aerodynamic improvements in terms of $C_p$. The twist
distribution might benefit from a refinement performed with a higher-fidelity aerodynamic model, which will be the subject of future work. As shown in Fig. 9, the structure of the optimal blade also undergoes a large upscaling, particularly in the spar caps. Manufacturing constraints to limit the thickness of these structural elements are available in the code, but were not used in the present exercise. Overall, the blade mass experiences a 77.9% growth.

    The tower also undergoes a significant upscaling, both due to aerodynamic advantages implied by a higher hub height and
because of the need to resist the higher loads produced by a larger rotor. The comparison between reference and optimal tower structures is shown in Fig. 10.

    It is interesting to notice that the monolithic structural optimization of rotor and tower structures performed by function `MinICC` (cf. Eq. (4)) finds a solution that shows a noticeable interaction between these two components. This is well visible on the left diagram in Fig. 10, where the distribution of outer diameters shows a step behavior, whose effect is to increase
the clearance between tower and blade tip. The algorithm is then able to reduce blade mass thanks to the relaxation of the tip deflection constraint, which results in savings in ICC. Notice that it is not a standard practice to simultaneously optimize rotor and tower, while apparently this might lead to savings due to the correct consideration of the mutual effects of the two components. Overall, tower height moves from 115.6 m to 134.9 m, with a tower mass increase of 43.5%.

    A cost analysis of the combined preliminary-detailed optimization process shows a significant growth of the ICC, equal to
14.3%. This results from the growth of rotor, drive-train, nacelle and tower costs, equal to 34.0%, 29.5% and 3.5%, respectively. However, the associated massive growth of the AEP, which passes from 48.8 GWh to 57.2 GWh, largely justifies the higher costs, resulting in a CoE reduction of 7.0%.




Clearly, transportation, logistics or other considerations might lead to different conclusions as to the actual optimal config-uration. However, these and other effects could be translated into cost items in the CoE models, and therefore they could be readily integrated in the present conceptual framework with no substantial difficulty.

The computational cost of the design optimization of the 10 MW wind turbine was larger than the one of the 2.2 MW,

possibly due to an initial guess farther away from the optimum. The final design was found in 6 iterations of the SQP algorithm, with a total computational time of approximately 100 hours on a workstation equipped with 40 logical processors. Even in this case, for the same reasons noted above, significant improvements to the computational time are possible.

### 3.2.2  Low induction configuration

A LI configuration is also investigated for the offshore 10 MW machine, using the same methodological approach used for the

2.2 MW case. A holistic optimization returned even in this case a traditional non-LI design.

To further investigate the concept, in a second attempt unexceedable loads from the blade root down to the rest of the wind turbine structure were assumed. Such a design solution could indeed be attractive in the context of a partial redesign effort, such as a reblading of the rotor, whereas a full redesign would require massive changes in terms of technologies, supply chain, manufacturing processes, logistics, etc. In such a situation, one could try to improve the CoE by increasing the rotor radius,

while at the same time not exceeding some of the loads of the baseline machine. This approach is performed with the proposed methodology by assuming a frozen wind turbine configuration except for the rotor radius, with $\boldsymbol{p}_g$ that reduces in this case to the following:

$$\boldsymbol{p}_{g_{LIR}} = [R, \delta]. \tag{10}$$

The baseline values for rotor thrust and blade root combined moment are selected as constraints for this partial redesign

effort. Although other choices are indeed possible, such a simple solution somewhat translates the requirements of not exceed-ing the baseline loads in the rest of the machine. Under these conditions, a optimal LIR design is found at a rotor diameter of 188.5 m, corresponding to a growth of 5.7% compared to the baseline design, and a pitch offset of 2.1 deg. The rotor shows a drop in $C_p$ equal to -0.4%, but an increase in AEP thanks to the larger rotor swept area of 2.8%. Overall, savings of about 2.0% are found in the CoE.

The main drawback of such an approach is that only loads that come from operational conditions in region II can be con-strained in a LIR, while storm, shutdown or loads generated in region III are not influenced by $\delta$ and may require a more careful assessment (Bottasso et al., 2014b). Table 11 reports a summary of the load analysis, indicating that some important loads do indeed come from conditions that are not affected by a LI design. In conclusions, the LIR configuration found this way may be attractive, but only when a lower induction can indeed reduce all driving loads in all components, a condition that is seldom if

ever verified.



## 4 Conclusions

This paper has presented integrated and high-fidelity design methodologies for wind turbines, that marry for the first time preliminary and detailed design procedures. The proposed algorithmic process aims at a minimization of the CoE merit figure at constant rated power. This is obtained by a novel procedure that stacks in sequence a rotor aerodynamic optimization for

maximum AEP and a monolithic rotor and tower structural optimization for minimum ICC. An external loop optimizes rotor radius, hub height, rotor cone angle, nacelle uptilt angle and the blade aero-structural configuration. Next, an aerodynamic sub-loop optimizes chord, twist and thickness distributions for a given choice of airfoils and for given aero-structural constraints on rotor shape. Lastly, a structural sub-loop identifies the optimal thickness distributions of the blade structural components, such as shell skin, spar caps, webs and reinforcements, and the optimal distributions of diameter and wall thickness along the

tower. The output of the procedure is the optimized design of a wind turbine, including details on blade shape, blade structure, tower structure, control parameters, load envelopes at all verification stations, as well as costs of the various components.

    This novel design methodology is applied to two reference wind turbine designs: a commercial-scale 2.2 MW onshore machine and a conceptual next-generation 10 MW offshore wind turbine. In the first case, the machine is found to be slightly under-designed in terms of rotor radius and hub height. Moreover, the blade aero-structural configuration is altered, by increas-

ing chord and thickness distributions. Improvements in the wind turbine design jointly improve the cost of the machine and the AEP, resulting in a CoE reduction of 3.1%. The redesign of the 10 MW wind turbine leads to more pronounced advantages in terms of CoE, as the size of the reference baseline machine is found to be significantly smaller than the optimum identified by the proposed procedures. Despite a massive increase in ICC, the larger rotor swept area and the higher average wind speed lead to a higher AEP that more than offsets the increase in cost, in turn leading to a CoE reduction of about 7.0%.

Overall, significant design changes are obtained for wind turbines that were already considered as very reasonable solutions. The new optima are identified in a completely automatic manner, by the integration of the preliminary-design-level macro parameters with detailed-design-level structural and aerodynamics variables. In addition, the monolithic optimization of rotor and tower, together with rotor cone and uptilt, is capable of finding best-compromise solutions through the couplings induced by the blade tip clearance constraint. Finally, thanks to the high-fidelity simulation and verification models used within this

framework, results are expected to be close to industrial products. As the optimization is conducted according to international standards, results should also be readily certifiable.

    In addition to the full design optimization, in this study LIR configurations are investigated to evaluate the potential benefits of a reduced induction coefficient and the potentially reduced associated loads. This capability is obtained by introducing an offset design variable to the rated pitch angle. The cases considered in the presented study show that LIR solutions do not

appear to be optimal, as standard optimal efficiency rotors appear to be in general associated with lower values of CoE. LIR optimal solutions were only obtained when constraining maximum loads on wind turbine components other than the blades. However such a result may only appear for machines that are not driven by loads generated during storms, shutdowns or other conditions when low induction does not help.



Ongoing work is proceeding on various fronts to further improve the methods by increasing their generality and level of sophistication. Among the various features under investigation, we mention here the ability to perform multi-objective and/or Pareto front optimizations, which are useful for generating families of optimal solutions instead of single points, as well as probabilistic optimization methods that can take into account uncertainties in data, operating conditions and models.

5 *Acknowledgements.* The present work is partially supported at the Politecnico di Milano by the EU FP7 INNWIND project.





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



**Table 1.** Configuration of the 2.2 MW onshore wind turbine.

| Data | Value | Data | Value |
|---|---|---|---|
| Wind class | IEC 3A | Rated mech. power | 2.2 MW |
| Hub height | 80.0 m | Rotor diameter | 92.0 m |
| Cut-in | 3 m/s | Cut-out | 25 m/s |
| Rotor cone | 2.0 deg | Nacelle uptilt | 6.0 deg |
| Rotor solidity | 4.65% | Max $V_{tip}$ | 72.0 m/s |
| Blade mass | 7482 kg | Tower mass | 119.2 ton |

**Table 2.** Spanwise positioning of the airfoils for the 2.2 MW onshore wind turbine.

| Airfoil | Position |
|---|---|
| Circle | 0.0% |
| Circle | 2.22% |
| DU00-W2-401 | 19.43% |
| DU00-W2-350 | 25.53% |
| DU97-W-300 | 35.04% |
| DU91-W2-250 | 47.69% |
| DU93-W-210 | 69.44% |
| DU95-W-180 | 89.22% |
| DU95-W-180 | 100.00% |




**Table 3.** Extent of the structural components and their materials for the 2.2 MW wind turbine.

| Component | From (% span) | To (% span) | Material type |
|---|---|---|---|
| External shell | 0 | 100 | Stitched triaxial -45°/0°/+45° fiberglass |
| Spar caps | 1 | 98 | Unidirectional fiberglass |
| Shear webs | 10 | 98 | Stitched biaxial -45°/+45° fiberglass |
| Trailing and leading edge reinforcements | 10 | 98 | Unidirectional fiberglass |
| Sandwich core | 5 | 98 | Balsa |

**Table 4.** Summary of the material properties used in the blades of the 2.2 MW wind turbine.

| Material type | Longitudinal Young's modulus [MPa] | Transversal Young's modulus [MPa] | Shear modulus [MPa] |
|---|---|---|---|
| Stitched triaxial fiberglass -45°/0°/+45° | 28544 | 10280 | 6470 |
| Unidirectional fiberglass | 39277 | 8450 | 3190 |
| Stitched biaxial fiberglass -45°/+45° | 9737 | 9737 | 10913 |
| Balsa | 50 | 50 | 150 |





**Table 5.** Summary of design parameters $\boldsymbol{p}_g$ for the holistic optimization of the 2.2 MW onshore wind turbine.

| Data | Reference | Optimum | Difference |
|---|---|---|---|
| Rated mech. power | 2.2 MW | 2.2 MW | – |
| Rotor diameter | 92.0 m | 106.6 m | +15.9% |
| Hub height | 80.0 m | 97.6 m | +22.0% |
| Rotor cone | 2.0 deg | 2.2 deg | +10.0% |
| Nacelle uptilt | 6.0 deg | 6.5 deg | +8.3% |
| Rotor solidity $\sigma_c$ | 4.64% | 4.26% | -8.2% |
| Blade tapering $\tau_c$ | 0.419 | 0.414 | -1.2% |
| Thickness solidity $\sigma_t$ | 0.342 | 0.348 | +1.8% |
| Thickness tapering $\tau_t$ | 0.344 | 0.362 | +5.2% |

**Table 6.** Configuration of the 10 MW offshore wind turbine.

| Data | Value | Data | Value |
|---|---|---|---|
| Wind class | IEC 1A | Rated mech. power | 10.6 MW |
| Hub height | 119.0 m | Rotor diameter | 178.3 m |
| Cut-in | 4 m/s | Cut-out | 25 m/s |
| Rotor cone | 4.65 deg | Nacelle uptilt | 5.0 deg |
| Rotor solidity | 4.66% | Max $V_{tip}$ | 90.0 m/s |
| Blade mass | 42496 kg | Tower mass | 628.0 ton |

**Table 7.** Spanwise positioning of the airfoils for the 10 MW wind turbine.

| Airfoil | Position |
|---|---|
| Circle | 0.0% |
| Circle | 1.74% |
| FFA-W3-480 | 20.80% |
| FFA-W3-360 | 29.24% |
| FFA-W3-301 | 38.76% |
| FFA-W3-241 | 71.87% |
| FFA-W3-241 | 100.00% |



**Table 8.** Extent of the structural components and their materials for the 10 MW wind turbine.

| Component | From (% span) | To (% span) | Material type |
|---|---|---|---|
| External shell | 0 | 100 | Stitched triaxial -45°/0°/+45° fiberglass |
| Spar caps | 1 | 99.8 | Unidirectional fiberglass |
| Shear webs | 5 | 99.8 | Stitched biaxial -45°/+45° fiberglass |
| Third shear web | 22 | 95 | Stitched biaxial -45°/+45° fiberglass |
| Trailing and leading edge reinforcements | 10 | 95 | Unidirectional fiberglass |
| Root reinforcement | 0 | 22 | Unidirectional fiberglass |
| Shell core | 5 | 99.8 | Balsa |
| Web core | 5 | 99.8 | Balsa |





**Table 9.** Summary of the material properties used in the blades of the 10 MW wind turbine.

| Material type | Longitudinal Young's modulus [MPa] | Transversal Young's modulus [MPa] | Shear modulus [MPa] |
|---|---|---|---|
| Stitched triaxial -45°/0°/+45° fiberglass | 21790 | 14670 | 9413 |
| Unidirectional fiberglass | 41630 | 14930 | 5047 |
| Stitched biaxial -45°/+45° fiberglass | 13920 | 13920 | 11500 |
| Balsa | 50 | 50 | 150 |

**Table 10.** Summary of design parameters $p_g$ for the holistic optimization of the 10 MW onshore wind turbine.

| Data | Reference | Optimum | Difference |
|---|---|---|---|
| Rated mech. power | 10.6 MW | 10.6 MW | – |
| Rotor diameter | 178.3 m | 223.2 m | +25.2% |
| Hub height | 119.0 m | 138.3 m | +16.2% |
| Rotor cone | 4.65 deg | 5.51 deg | +18.5% |
| Nacelle uptilt | 5.00 deg | 5.25 deg | +5.0% |
| Rotor solidity $\sigma_c$ | 4.66% | 4.08% | -12.4% |
| Blade tapering $\tau_c$ | 0.429 | 0.406 | -5.4% |
| Thickness solidity $\sigma_t$ | 0.389 | 0.389 | +0.0% |
| Thickness tapering $\tau_t$ | 0.358 | 0.358 | +0.0% |



**Table 11.** Summary of load analysis for the LIR design of the 10 MW onshore wind turbine.

| Load | Driving DLC baseline | Driving DLC optimum | Load difference |
|------|----------------------|---------------------|-----------------|
| Blade root combined moment | DLC13 @ 13 m/s | DLC13 @ 13 m/s | -0.3% |
| Blade root torsional moment | DLC62 @ 30degYM | DLC62 @ 30degYM | +1.8% |
| Rotor thrust | DLC13 @ 13m/s | DLC13 @ 13m/s | -5.2% |
| Tower base combined moment | DLC62 @ -30degYM | DLC62 @ 60degYM | +14.8% |
| Hub overturning moment | DLC13 @ 23 m/s | DLC13 @ 25 m/s | +18.0% |
| Yaw bearing moment | DLC13 @ 17 m/s | DLC62 @ 60degYM | +21.9% |





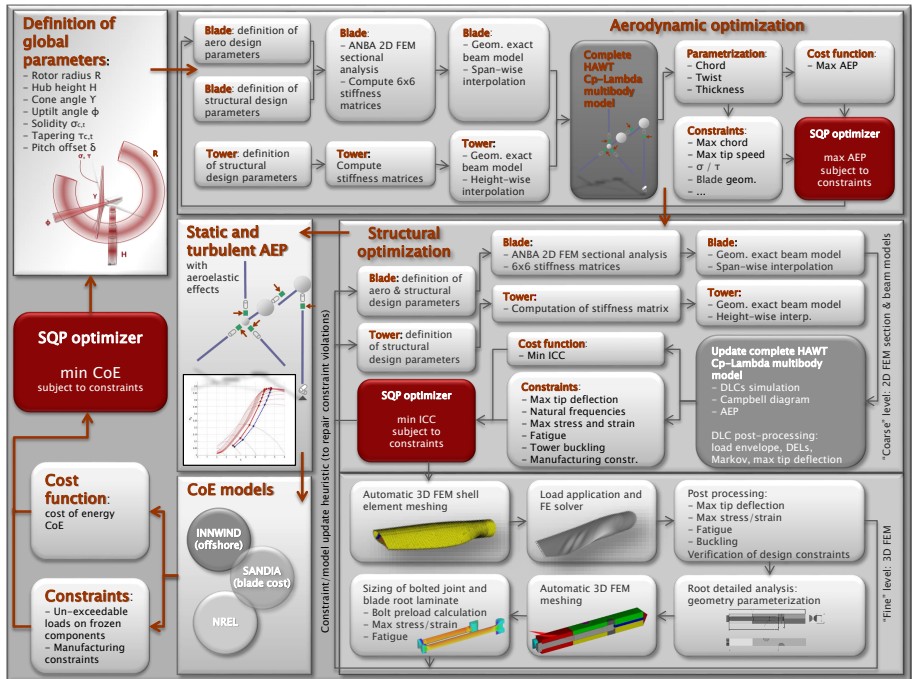

**Figure 1.** Overall architecture of the multi-level combined preliminary-detailed design procedure.

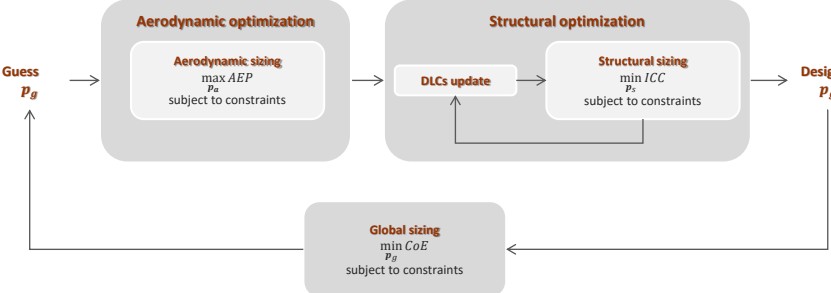

**Figure 2.** Architecture of the combined preliminary-detailed design procedure.




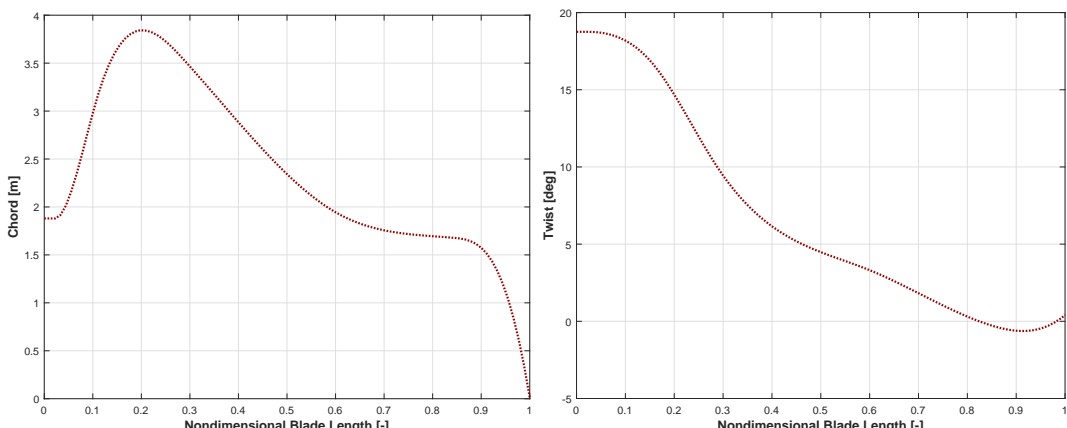

**Figure 3.** Baseline chord and twist distributions for the 2.2 MW wind turbine blade.

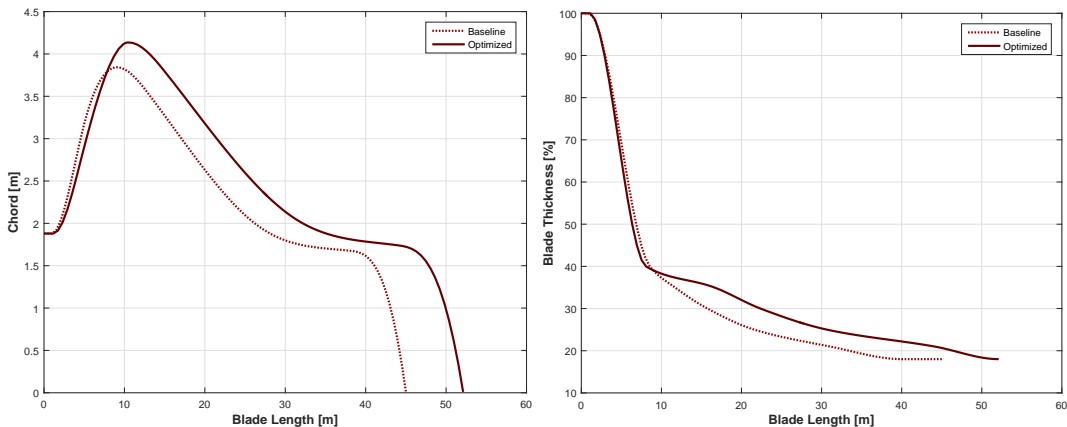

**Figure 4.** Chord and thickness distributions of the baseline and the optimized 2.2 MW wind turbine blades.




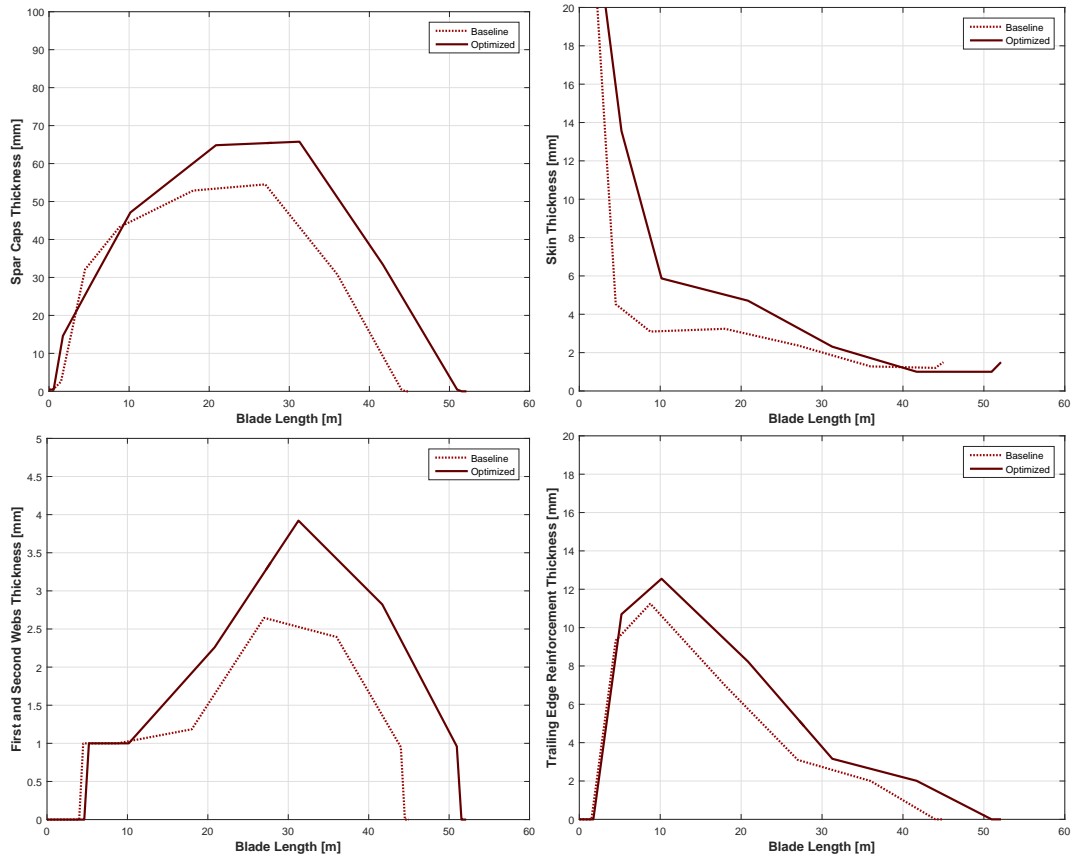

**Figure 5.** Structural thickness distributions of the baseline and the optimized 2.2 MW wind turbine blades.

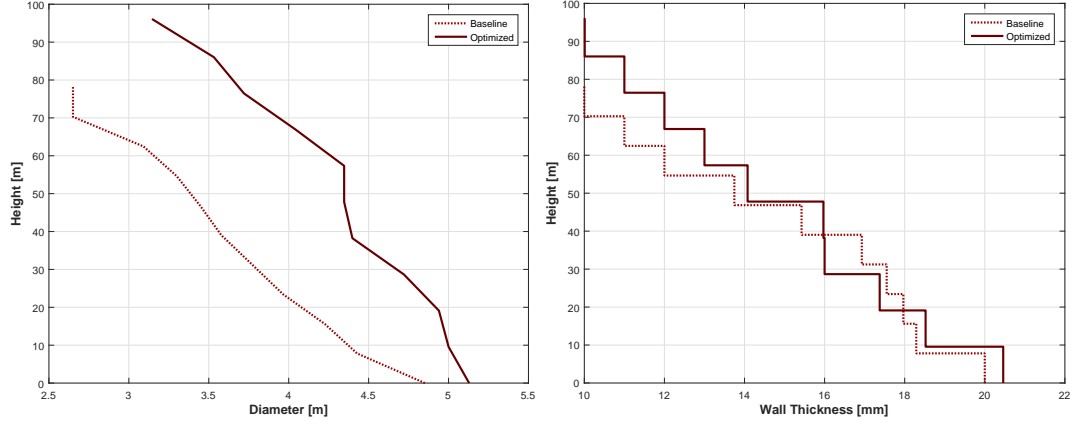

**Figure 6.** Tower outer diameter and wall thickness distributions of the baseline and the optimized 2.2 MW wind turbines.




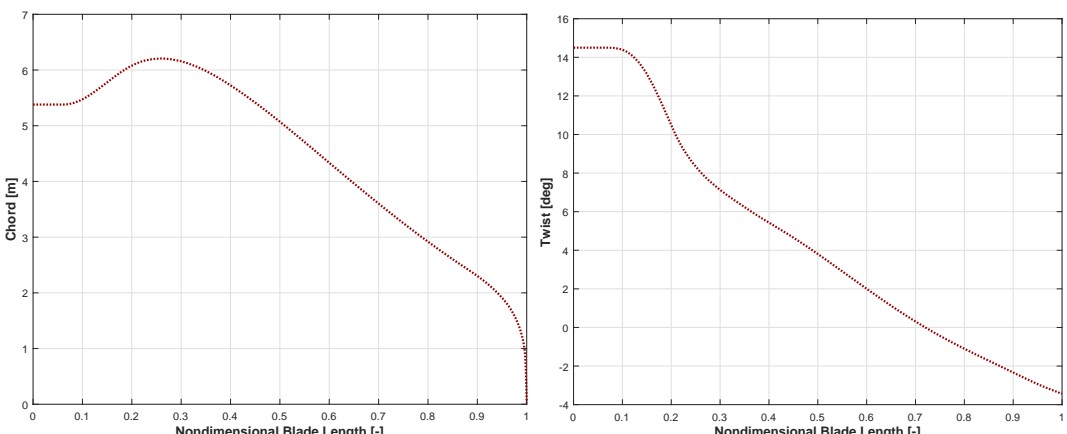

**Figure 7.** Baseline chord and twist distribution for the 10 MW wind turbine blade.

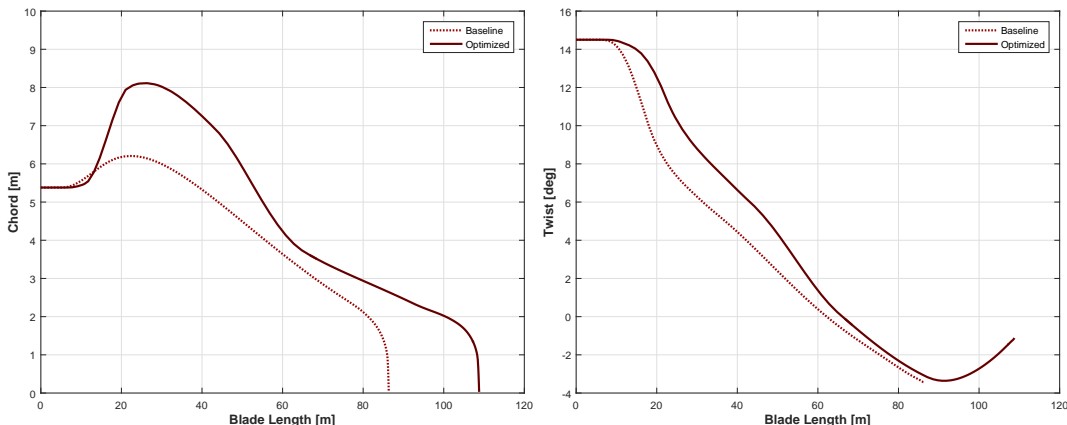

**Figure 8.** Chord and thickness distributions of the baseline and the optimized 10 MW wind turbine blades.




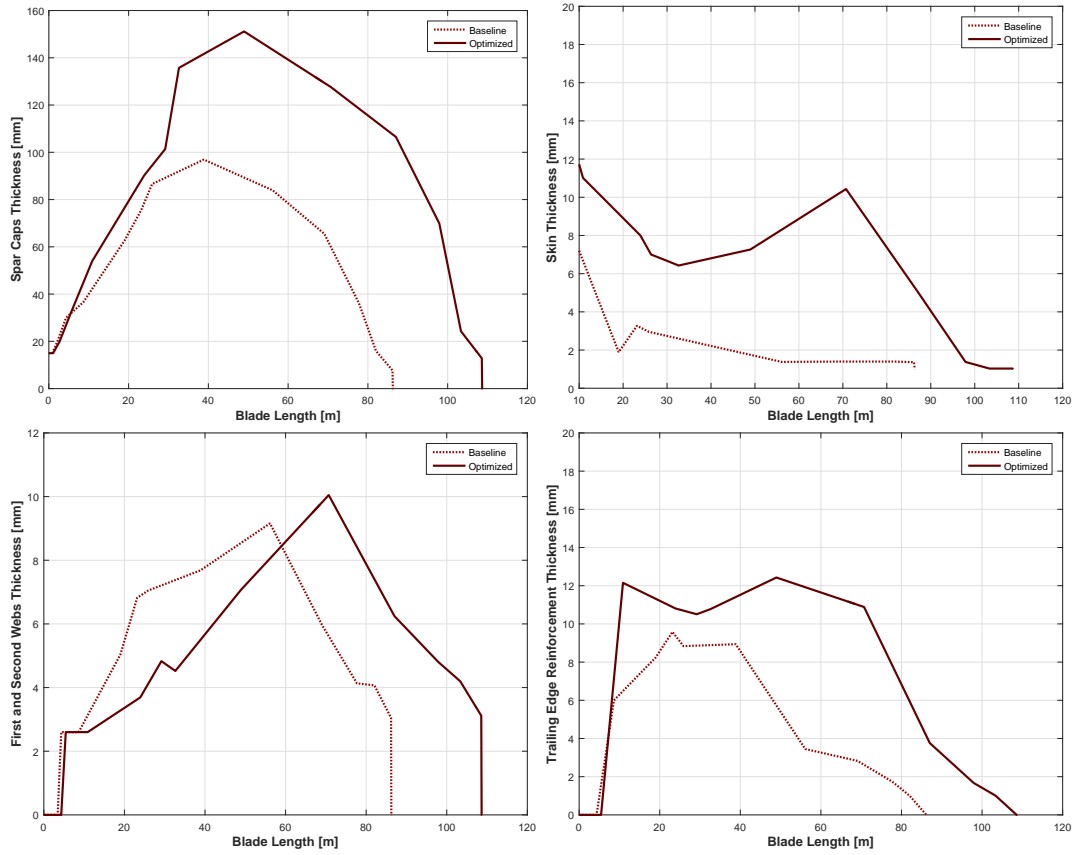

**Figure 9.** Structural thickness distributions of the baseline and the optimized 10 MW wind turbine blades.

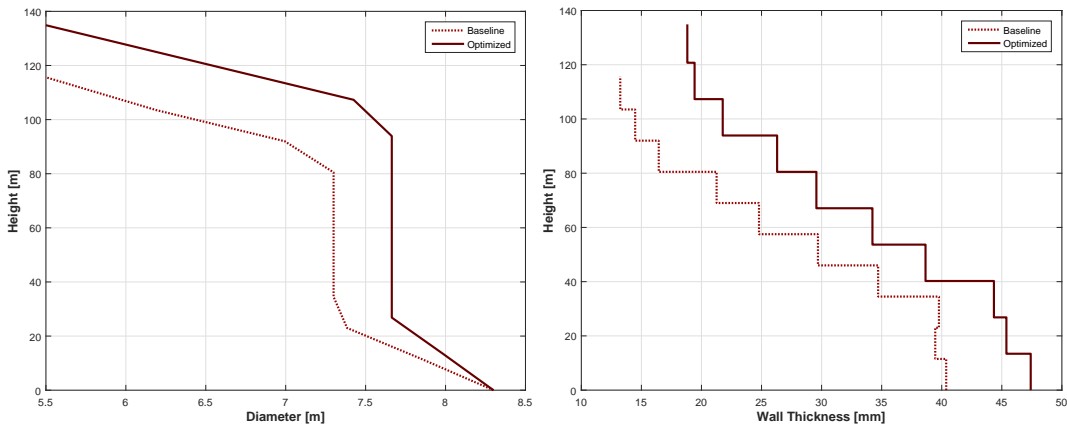

**Figure 10.** Tower outer diameter and wall thickness distributions of the baseline and the optimized 10 MW wind turbines.