# Peer review of "Combined preliminary-detailed design of wind turbines"

_Wind Energy Science, 2015_

## Referee Comment (RC1) · Anonymous Referee #1 · 25 Feb 2016

Subject: Review of manuscript submitted to Wind Energy Sci. Title of the paper: Combined preliminary-detailed design of wind turbines Authors: P. Bortolotti, C. Bottasso, A. Croce

The manuscript under review presents a combined preliminary-detailed optimization method to do the sizing of structural and aerodynamic design of wind turbines using sequential optimization. Rotor and tower of the wind turbine are designed simultaneously with stresses, buckling, blade-tower tip clearance, modal frequencies and fatigue as the design constraints. The objective function is to minimize the cost of energy. The method is applied to a 2.2 MW onshore, and a 10 MW offshore wind turbine. Results show some improvements in the COE with respect to the initial design.

The manuscript is well written in English, but its structure and level of details are not

well defined. Also the value proposition of the paper is not clearly defined. My recommendation is to accept the paper but with major revision. Below are some comments for the authors to fully revise their paper:

Major comments: Abstract section: "A new procedure is introduced that marries ...". This is a strong statement to say. First of all, it is nowadays a common industrial practice to do this type of analysis, though the wind turbine manufacturers do not talk about it publicly. Second, earlier papers have up to some extent the same methodology as presented here. For instance: - Kenway GKW, Martins JRRA. Aerostructural Shape Optimization of Wind Turbine Blades Considering Site-Specific Winds. In: Proceedings of the 12th AIAA/ISSMO Multidisciplinary Analysis and Optimization Conference. Proceedings of the 12th AIAA/ISSMO Multidisciplinary Analysis and Optimization Conference. Victoria, BC; 2008. AIAA 2008-6025

Finally, there are several wind turbine computational codes that have the same capability as presented in this paper. PHATAS-FOCUS from ECN and the S4WT are among codes that can do this type of analysis and optimization. My suggestion is to revise this statement. The merit of this paper will not be judged by this claim that may or may not be true.

Why the authors believe that the combined preliminary-detailed design is important? What is the advantage of this approach compared to other design techniques? Please show how this technique can solve an actual problem. For that, the authors need to find a knowledge gap in the existing design procedure that the proposed approach can solve. You need to explain to the reader clearly why is this a good approach in terms of its value proposition.

Nothing is said in the paper on how the 3D turbulent wind is generated and used? What about sheer and veer in the wind? Any coherent directional change in the wind? What about other aspects of the flow field modeling like the dynamic stall, dynamic inflow and 3D effects? Some of these issues are particularly important when the optimization

happens. As an example, the 3D effects have dependency on the blade aspect ratio that changes in every optimization iteration. Please clarify.

Nothing is said on how the controller works in this approach? Please explain how the authors did the optimization of the rotor, while in every optimization iteration the same controller may not be representative? If it changes, then how?

While changing the rotor diameter, the rated rotor speed and wind speed also change. This influences the controller design switching algorithm. How is this seen in the existing setup?

It is not clear what the safety factors are for the initial design and the optimized design of the two wind turbines. One can always do an optimization and provide a better optimal by reducing the safety factors. Please comment on this, since this is important.

What happened to the hydrodynamic loads of the 10 MW offshore wind turbine? Is there any hydro loading considered? Is the optimization of this design without considering and hydroelasticity in place?

Page 11, line 31: Doing an optimization with only one seed is not realistic. What if the seed that you used gives much smaller loads on the design than the original design. In this case you can always claim to optimize the design which is not necessarily fair to claim. Please do a multi-seed analysis and average, max, etc the outcomes.

How do the thickness of the composite layups change? Do they change continuously (which is not realistic) or they are introduced as discrete variables? Is the number of laminates fixed? What about the angels? Please explain.

So no transportation, logistics and installation in the cost models, but what if the 10 MW turbine has optimized the design considering these issues, and you not? You can make a design better than what they made since you are relaxing the design space allowing to search for a better optimum and claim that the 10 MW is not optimal. Please explain.

First line of conclusion: "This paper presents an integrated ...". This is integrated but

a sequential design. Also only the structural design is high-fidelity and not the aerodynamic design. Please be more precise.

Page 16, line 25: The design is far from the industrial practice. A single seed, limited DLCs, no soil-structure interaction, a frozen controller parameters, etc. I think it is better not to have such statements in the paper, since the authors do not know what industry is exactly doing, and this leads to confusion for the readers.

I was not able to understand how the authors considered ground-blade clearance? Is there any limit on how close it can gets to the ground? Additionally, it seems that there is a strong wind shear present since the optimizer is trying to increase the hub-height beyond what the optimal is. Please explain.

Minor comments: Section 2.2.1: How is the AEP computed? Is the Weibull distribution function considered? If so, what are the scale and shape factors?

Equation 2: This type of presentation is difficult to read from a readers point of view. My suggestion is to replace it with a figure where the data and process flow can be presented visually.

Page 10, line 2: What is the approach for computing the gradients for the optimization? Finite difference? Forward, backward, central?

Please provide a table with all the DLCs and their corresponding parameters in details.

Please provide a figure to allow comparing both the flapwise and edgewise stiffnesses of the two designs? It is not clear how globally these important design properties change with blade length for the original and the optimal designs? The same thing with mass distribution.

Please provide a comparative table to show the cost and mass of the optimal and original designs, as well as the AEP and COE. In this way, once can see what happens to the wind turbine during the optimization, and what changes and what not.

Page 13, last line: "higher than 1.1 drives ...". I did not understand what the authors want to say.

How many discrete stations are used along the blade and tower to do the optimization for both the design variables and design constraints?

I do not understand. Why the authors want to say something in the middle of the paper on low induction concept. This is a side activity and distracts the reader from focusing on the framework. Please consider removing that and focusing more on the details of the method.

Page 16, line 11: Nothing is said about the controller, and controller parameters.

Please provide the modal frequencies of the original and optimized design for the purpose of comparison. Additionally, how are these constraints defined and what is their lower and upper limits?

Please provide the details on how the fatigue damage calculation is done? What are the parameters, properties, and lower and upper limits defined as a design constraints?

Similar to fatigue damage and frequencies, the details of other constraints, the way they are calculated and the upper and lower limits of each of them and how they are satisfied in the design are missing. Please provide these details.

Overall, the paper misses lots of technical details and some of the claims made by the authors is not on a fair basis to compare with the original design in terms of the improvements. Also no supporting arguments is provided to believe that these claims are correct. This particularly applies to the 10 MW offshore design. The paper needs a major revision before it can go to the next step in the publication process.

---

## Referee Comment (RC2) · P. Veers (Referee) · 7 Mar 2016

Review of Combined preliminary-detailed design of wind turbines

**Summary**

The system wide optimization of wind turbines represents a step change in the level of sophistication in system optimization, moving beyond the previous generation where the aerodynamics were optimized first and the structure and manufacturing were left to be optimized within the constraints produced by the aerodynamic and production requirements. The ability to optimize structural and manufacturing in conjunction with aerodynamic performance has led to the current generation of turbines with larger rotors and aerodynamic performance sometimes off optimum to allow for more efficient structural designs. This paper takes that current practice and expands the envelope to consider tower design, height, and diameter in the optimization loop. Although it claims to be the first, it is at the leading edge of an emerging area of work in which there have already been forays into these combinations of design variables (e.g., Ning, A.; Dykes, K. (2014), see below). This work arguably brings this work to a significantly higher level of fidelity in the modeling of individual subsystems within the turbine system. It is expected that future work in this areas will continue beyond the individual turbine design and into the realm of full wind plant optimization were each turbine is allowed to optimally fit its particular location and operating constraints.

This paper is silent in the area of how the drive train component sizing if affected by design elements other than rotor torque and rotor weight. Since the design standards make no such connection, it is understandable that this optimization illustration does not attempt to create and then implement linkages to imbalanced rotor loads, but it would supply another restraining factor to the growth to much larger rotors. This is also an area of potential future work, as is a connection to mechanical subsystem reliability. Both of these are significant additions to this work and their absence does not diminish the accomplishment at all.

The demonstration of this new optimization capability and the ensuing benefits is the main outcome of this work, but it supplies many other useful results that might be highlighted in the article. First, the optimized version of the baseline machines provides something of an improved baseline in each case. The closer a baseline design is to the global optimum for the embedded technology, the more useful it becomes as a measure against with new approaches and technologies can be measured. These altered configurations could be considered as the baseline, Rev. 2. There have been a number of studies proposing technology innovations where the use of a suboptimal baseline makes the change to new technology almost guaranteed to produce an improved outcome. The publication of improved optimal baselines would be a great benefit if used in these studies.

Second, the work has explored the design space at two interesting turbine sized and has exercised design constraints that differ at each size. These design constraints themselves are very interesting outcomes of the study and a more explicitly display of them would be very useful. While those active in commercial design may know what drives design, many researchers are not knowledgeable, and publish results assuming great impact when in fact the studied impact is not design driving at all. This paper could provide guidance in this area to a great many researchers.

The study also examines the use of Lower Induction (LI) designs to reduce structural loads on the rotor and find that there is little to gain in the land-based turbine, but the 10MW offshore machine could benefit by also growing the rotor and increasing swept area. The LI is achieved with a pitch offset. It would be interesting to know how a change in pitch (twist) schedule along the blade to reduce induction in a more tailored way might enhance the outcome of that option. Perhaps some discussion of the point could be included by the authors.

The authors do an excellent job of providing context and conditions that help the readers with their expectations of how generally useful this particular study should be. They note limitations that might be imposed by logistics, and the scattered definition of design load cases and storm loads might otherwise drive a specific design. This is a very readable and informative article. However, there could be a more direct statement of the importance of the cost models in driving the optimization result, as well as the analysis capabilities, which are the main contribution the authors have brought to this area. Cost models and assumed step changes in cost elements at points where manufacturing or transportation constraints come into play should be highlighted to a greater extent than currently stated. Overall, this is an excellent study.

Ning, A.; Dykes, K. (2014). "Understanding the Benefits and Limitations of Increasing Maximum Rotor Tip Speed for Utility-Scale Wind Turbines," Article No. 012087. *Journal of Physics: Conference Series.* Vol. 524(1), 2014; 10 pp.; NREL Report No. JA-5000-61729.)

**Issues for Consideration – Mandatory changes**

- 1. Revise the statements the statements that this is the first time optimization has included tower and rotor size to be more consistent with the current literature (e.g., see above)
- 2. I don't recall seeing a mention of any speed constraint on the rotor. Was there a tip speed constraint and how was it determined?
- 3. Discuss the issue of rotor non-torque loads and how they were or were not included in the costing of the drive train components. It is assumed the drive train was designed for the steady loads, but it just needs to be made clear.
- 4. In two places the term "under-designed" is used. It appears to mean under sized, but it is not clear. A more clear term should be used.
- 5. Plots such as Figure 6 showing blade design parameters should be labeled as "Blade span" instead of "Blade Length"
- 6. Plots such as Figure 5 showing tower parameters should have the independent variable of height on the x-axis and the dependent variables (such as thickness and diameter) on the y-axis to be consistent with the blade results.

Issues for Consideration – Optional changes

1. The practice of using software language statements (subroutine names) in the equations documenting the optimization process seems to be growing, especially in this area of system optimization. While it may provide better understanding to the practitioners of this area of

expertise, it is not as clear to those (like this reviewer) who are outside the area. Consider revising the equations to use more traditional mathematical symbols and descriptors.

- 2. On page 5, the first paragraph, there is a statement that no dependence of H is assumed for the 50 year storm wind speed. That sound ominous and could use more explanation.
- 3. Better define the term "un-exceedable loads".
- 4. Discuss the option to introduce the LI designs through a distributed change in blade pitch (twist) rather than through a single offset.
- 5. On page 7 it would be useful to describe each IEC DLC briefly rather than just referring to each number.
- 6. Also on p. 7, explain how close the "close to be active" the frequency constraints are.
- The design driving load cases that provided active constraints are very nicely listed and discussed. Consider creating a table that summarizes them for each design size. These are very interesting results.

---

## Author Comment (AC1) · 23 Mar 2016

**Combined Preliminary-Detailed Design of Wind Turbines**

**Detailed Replies to Reviewers**

**Reviewer #1**

We thank the reviewer for the detailed analysis of our work and the long list of inputs, comments and suggested improvements. The revised version of the manuscript is included at the end of this document, with a blue highlight to indicate all changes with respect to the previous version. A list of point-by-point replies to the reviewer's comments is reported in the following:

1. [**Reviewer**] *Abstract section: "A new procedure is introduced that marries ...". This is a strong statement to say. First of all, it is nowadays a common industrial practice to do this type of analysis, though the wind turbine manufacturers do not talk about it publicly. Second, earlier papers have up to some extent the same methodology as presented here. For instance: - Kenway GKW, Martins JRRA. Aerostructural Shape Optimization of Wind Turbine Blades Considering Site-Specific Winds. In: Proceedings of the 12th AIAA/ISSMO Multidisciplinary Analysis and Optimization Conference. Proceedings of the 12th AIAA/ISSMO Multidisciplinary Analysis and Optimization Conference. Victoria, BC; 2008. AIAA 2008-6025*
   *Finally, there are several wind turbine computational codes that have the same capability as presented in this paper. PHATAS-FOCUS from ECN and the S4WT are among codes that can do this type of analysis and optimization. My suggestion is to revise this statement. The merit of this paper will not be judged by this claim that may or may not be true.*

   [**Authors**] We have now revised this sentence, removing the statement about the complete novelty of the approach.

   However, we do not entirely agree with the observations of the reviewer. In fact, based on our current and past industrial collaborations and quite extensive consulting activities, we do not think that multi-disciplinary design codes are commonly adopted by industry, possibly with extremely few exceptions. Even if they were, this knowledge is not in the public domain because it has not been published. Therefore, in our opinion, there is a need for papers, as the present one, that describe methods for the multi-disciplinary design of wind turbines through peer-reviewed publications. Saying that something has been done already, although nobody knows how because it has never been disclosed publicly, is not a constructive contribution to the creation of knowledge within the scientific and technical communities.

   Regarding the AIAA paper mentioned by the reviewer, that work does not include an optimization of the macro parameters of a wind turbine, such as rotor radius or hub height, as we do in this paper –and which is in fact one of the key novelty aspects of our contribution.

   Regarding the FOCUS 6 or S4WT packages, to our knowledge, this same capability is also not part of these codes, which in addition have not been the subject of any peer reviewed publication.

2. *[**Reviewer**] Why the authors believe that the combined preliminary-detailed design is important? What is the advantage of this approach compared to other design techniques? Please show how this technique can solve an actual problem. For that, the authors need to find a knowledge gap in the existing design procedure that the proposed approach can solve. You need to explain to the reader clearly why is this a good approach in terms of its value proposition.*

[**Authors**] The aim of the whole paper is exactly to demonstrate that the combined preliminary-detailed design is important. Our opinion is that this is clearly shown by the evident advantages in terms of CoE that were obtained in the examples of our paper for machines that had already been considered as optimal. Moreover, these methodologies help in clarifying the potential of new configurations, such as the low induction rotors studied in our paper. In addition, as clearly stated in the paper, it is quite evident that in principle all optimization procedures could be realized also by hand through a laborious sequential improvement of the design. However, it should also be self-evident that an automated way of conducting the analysis has its own advantages, and therefore deserves to be described.

3. *[**Reviewer**] Nothing is said in the paper on how the 3D turbulent wind is generated and used? What about sheer and veer in the wind? Any coherent directional change in the wind? What about other aspects of the flow field modeling like the dynamic stall, dynamic inflow and 3D effects? Some of these issues are particularly important when the optimization happens. As an example, the 3D effects have dependency on the blade aspect ratio that changes in every optimization iteration. Please clarify.*

[**Authors**] We use TurbSim for the turbulent wind time histories and our own software to generate deterministic gusts, always in exact compliance with IEC standards. Text has been added in §2.1 to mention TurbSim and the corresponding reference.

The coherent directional change was not found to be among the design drivers of the baseline machines and it was therefore not included in the list of DLCs considered in the study, as written in §3.1 and 3.2. We agree that in general this load case should be included, but its exclusion in this specific case not only does not affect the results –as it does not generate design driving loads-, but also evidently does not limit at all the overall algorithmic structure of our method, which is the methodological contribution of this paper. In other words, adding or removing DLCs will affect the results but will not change the method.

Dynamic stall, inflow models, tip and hub losses etc. are all available features of Cp-Lambda, as stated in §2.1. More in general, as this is the fifth paper in a series of works on Cp-Max –all duly referenced here-, we avoided giving excessive details to limit the paper size and avoid losing focus. We have paid great attention in our previous papers to give as many details as possible on the numerous sub-modules that compose Cp-Max. In addition, we have numerous publications that give detailed descriptions of the aeroservoelastic simulator Cp-Lambda, including its unique flexible formulation and numerical algorithms. Going back to the previous mention made by the reviewer of other design codes, we believe that our code distinguishes itself -at least- for the great attention that we have paid to the publication and dissemination of our methods and tools. In this contribution, we focused on some novel aspects (namely, the integration of macro parameters with detailed sizing in the overall design process), and therefore we felt that giving again a full description of all

aspects of our models would not only be out of scope, but would also be distracting from the actual focus of this paper.

4. [**Reviewer**] *Nothing is said on how the controller works in this approach? Please explain how the authors did the optimization of the rotor, while in every optimization iteration the same controller may not be representative? If it changes, then how? While changing the rotor diameter, the rated rotor speed and wind speed also change. This influences the controller design switching algorithm. How is this seen in the existing setup?*

[**Authors**] This is an important point that was left out in an attempt to reduce the paper size and improve text readability. Text has now been added to §2.1 and 2.2.2 to correct for this. The controller is a model-based LQR controller, described in detail in Bottasso et al, 2012 (now added to the bibliography). Being based on a reduced order model of the wind turbine, this formulation allows for the controller gains to be automatically updated whenever the wind turbine model changes.

5. [**Reviewer**] *It is not clear what the safety factors are for the initial design and the optimized design of the two wind turbines. One can always do an optimization and provide a better optimal by reducing the safety factors. Please comment on this, since this is important.*

[**Authors**] Safety factors are set by the standards, which we strictly adhere to. It is clear that a change in these factors would lead to an unfair comparison.

6. [**Reviewer**] *What happened to the hydrodynamic loads of the 10 MW offshore wind turbine? Is there any hydro loading considered? Is the optimization of this design without considering and hydroelasticity in place?*

[**Authors**] The 10 MW offshore wind turbine developed within the INNWIND project does not include hydrodynamic models. We have followed the same approach in this work to allow for a more direct comparison. Future activities may investigate the effects of hydrodynamic loading on the optimum design of this wind turbine. However, these changes will mostly involve the addition of hydrodynamic models, specific DLCs and modifications to the cost models, which again will certainly affect the results but not the optimization algorithms, which are the focus of this contribution.

7. [**Reviewer**] *Page 11, line 31: Doing an optimization with only one seed is not realistic. What if the seed that you used gives much smaller loads on the design than the original design. In this case you can always claim to optimize the design which is not necessarily fair to claim. Please do a multi-seed analysis and average, max, etc the outcomes.*

[**Authors**] The primary focus of this work is on the algorithms used for the optimization of wind turbines, and these are not influenced by the number of DLCs assumed in the study. A longer list in Eq. 3 would only enlarge Eq. 4c. We fully agree with the reviewer that during a real certification process a larger number of seeds should be included and this comment is added to §3.1 and in the conclusions.

8. [**Reviewer**] *How do the thickness of the composite layups change? Do they change continuously (which is not realistic) or they are introduced as discrete variables? Is the number of laminates fixed? What about the angels? Please explain.*

[**Authors**] The optimization is done internally by using continuous thickness variables, as the direct use of integers would lead to algorithmic complications (mixed integer-continuous variables). The continuous variables are however translated into discrete ones at every macro iteration that updates the loads by re-running all DLCs. During this update, all beam properties (inertial and structural) are updated too, using the discrete number of plies in the laminates. In the specific cases analyzed in this work, the difference in overall inertial and structural properties between discrete and continues variables is however very small, because of the substantial thicknesses involved. Text has been added in §2.2.2 to better explain this capability of the code.

As explained in detail in our referenced preceding papers, anisotropic laminated composites are modeled with a cross-sectional FEM procedure, which produces fully coupled 6x6 stiffness matrices. This allows, for example, to orient unidirectional laminates with given desired angles with respect to the pitch axis in order to achieve a bend-twist coupling behavior in the blades. However, fiber angles per se are not at present treated as optimization variables. Text has been included in §2.2.2, 3.1 and 3.2 to better explain this aspect of the models.

9. [**Reviewer**] *So no transportation, logistics and installation in the cost models, but what if the 10 MW turbine has optimized the design considering these issues, and you not? You can make a design better than what they made since you are relaxing the design space allowing to search for a better optimum and claim that the 10 MW is not optimal. Please explain.*

    [**Authors**] Transportation, logistics and installation costs are included in both the NREL and the INNWIND cost of energy models. To the authors' knowledge, the 10 MW machine was not truly optimized on these costs, but it was obtained as an upscaling of the 5 MW NREL wind turbine (cf. Bak et al. (2013)).

    It is anyway important to underline the importance of detailed cost models, and this point has now been better stated in §2.3.

10. [**Reviewer**] *First line of conclusion: "This paper presents an integrated ...". This is integrated but a sequential design. Also only the structural design is high-fidelity and not the aerodynamic design. Please be more precise.*

    [**Authors**] We disagree that it is a sequential design: the design is integrated in the sense that changes in any one discipline (aerodynamics, structures, controls) influences the others. The overall algorithmic flow is made of sequential steps, but this does not change the overall integrated nature of the approach.

    The high-fidelity term is clearly relative, but given the non-suitability of 3D CFD tools in a wind turbine rotor design context due to the extreme computational costs, we believe that the set of tools used in this paper is still of a relatively highly fidelity. For example, with reference to the previous comments of the reviewer on the current industrial practice, it is well known that major wind turbine designers use modal-based (therefore, linearized) structural models, without shear and very often also without torsional effects. Our approach (geometrically exact fully non-linear kinematics, 6 by 6 fully populated matrices, axial-shear-bending-torsion beam models) is certainly state-of-the-art with respect to this modeling aspect. In any case, we certainly agree with the suggestion of including more detailed aerodynamic analyses in future versions of the code, and this has now been added to the text.

11. [**Reviewer**] *Page 16, line 25: The design is far from the industrial practice. A single seed, limited DLCs, no soil-structure interaction, a frozen controller parameters, etc. I think it is better not to have such statements in the paper, since the authors do not know what industry is exactly doing, and this leads to confusion for the readers.*

[**Authors**] We do not agree on this statement. The single seed and the reduced list of DLCs are assumptions used in this study to validate the approach, but can be trivially removed in a real design context, with the sole effect of a higher computational cost. On the opposite, the soil-structure interaction and the update of the controller parameters are included in the study. In any case, text has been changed in multiple places within the document to take into accounts the comments of the reviewer.

12. [**Reviewer**] *I was not able to understand how the authors considered ground-blade clearance? Is there any limit on how close it can gets to the ground? Additionally, it seems that there is a strong wind shear present since the optimizer is trying to increase the hub-height beyond what the optimal is. Please explain.*

[**Authors**] We are not aware of any limitation on the ground-blade tip clearance prescribed by international standards. Therefore, we did not implement any constraint of such kind for the combination of rotor radius and hub height. This could anyhow be trivially implemented in case of need, and it could be included in the global constraints expressed by Eq. 5c. The explanation of the constraint is now included in §2.2.3.

Wind shear exponent is assumed to be 0.2 in compliance with the IEC guidelines as already mentioned in §2.2.3.

Answers to the reviewer's **minor comments** are reported in the following:

1. [**Reviewer**] *Section 2.2.1: How is the AEP computed? Is the Weibull distribution function considered? If so, what are the scale and shape factors?*

[**Authors**] The aerodynamic optimization is extensively explained in Bottasso et al., 2015 and, for sake of paper readability, only rapidly recalled here. A short explanation about AEP computation is however useful and it was now added to the paragraph.

2. [**Reviewer**] *Equation 2: This type of presentation is difficult to read from a readers point of view. My suggestion is to replace it with a figure where the data and process flow can be presented visually.*

[**Authors**] We have adopted this presentation format for the sake of a more formal, precise and complete description of the algorithms. The equation structure and symbolism, completely coherent with our previous publications on design optimization of wind turbines (Bottasso et al, 2011, Bottasso et al, 2013, Bottasso et al., 2014a, Bottasso et al., 2015), was designed to give readers a better understanding of the details of the (often quite involved) computational steps. In our opinion, this would not be possible only from text explanations, which would be extremely verbose and still probably not as precise. To support our text and formalism, we have also provided when possible a visual representation of the equations, as shown in Figures 1 and 2. Finally, we are not aware of an alternative clear and concise method to express these complex algorithmic structures.

3. *[**Reviewer**] Page 10, line 2: What is the approach for computing the gradients for the optimization? Finite difference? Forward, backward, central?*

[**Authors**] The gradients were calculated by finite differences, typically computed in the forward direction in the aerodynamic and structural loops and with a centered stencil in the global sizing. Text is modified in §2.2.1, 2.2.2 and 2.2.3 to include this information.

4. *[**Reviewer**] Please provide a table with all the DLCs and their corresponding parameters in details.*

[**Authors**] The list of DLCs is already included in the text together with a brief description of each load case. Interested readers can refer to the IEC standards, which are listed among the references.

5. *[**Reviewer**] Please provide a figure to allow comparing both the flapwise and edgewise stiffnesses of the two designs? It is not clear how globally these important design properties change with blade length for the original and the optimal designs? The same thing with mass distribution.*

[**Authors**] Flapwise and edgewise stiffness is, among a multitude of other information, output data that describes the resulting design, but it is not the subject of direct optimization. Being a design consequence -which descends from the material properties, topology and geometry of the various components-, we do not find these plots strictly necessary. However, we can certainly include these quantities if the associate editor finds them necessary for the discussion and not affecting the readability of the paper.

6. *[**Reviewer**] Please provide a comparative table to show the cost and mass of the optimal and original designs, as well as the AEP and COE. In this way, once can see what happens to the wind turbine during the optimization, and what changes and what not.*

[**Authors**] This information is already included in the text of §3.1.1 and 3.2.1. When the associate editor finds this part not clearly exposed, this information can be moved to tables.

7. *[**Reviewer**] Page 13, last line: "higher than 1.1 drives ...". I did not understand what the authors want to say.*

[**Authors**] The blades are subjected to a constraint that imposes the ratio of the first edgewise and flap eigenfrequencies of the blade to be higher than 1.1. The constraint is imposed by international standards to avoid the coalescence of these two modes. We have observed this constraint to be typically active, driving the trailing and leading edge reinforcements. Text was adjusted to facilitate the reader's understanding of this part of the paper.

8. *[**Reviewer**] How many discrete stations are used along the blade and tower to do the optimization for both the design variables and design constraints?*

[**Authors**] This information was added for both the 2.2 MW and the 10 MW cases.

9. *[**Reviewer**] I do not understand. Why the authors want to say something in the middle of the paper on low induction concept. This is a side activity and distracts the reader*

*from focusing on the framework. Please consider removing that and focusing more on the details of the method.*

[**Authors**] The goal of the paper is indeed to develop design methodologies, and we believe that the LIR study is an interesting alternative application of the same methods. The fact that LIR rotors may look beneficial in a low fidelity environment (cf. the referenced papers), but are found to produce no advantage in terms of CoE in a higher fidelity framework is a strong point in favor of the approaches that we advocate in this work. This could also help answering the reviewer's earlier doubts on the advantages of integrated combined preliminary-detailed design methodologies.

10. [***Reviewer***] *Page 16, line 11: Nothing is said about the controller, and controller parameters.*

[**Authors**] The controller data is automatically updated by the code, as now better explained in §2.1.

11. [***Reviewer***] *Please provide the modal frequencies of the original and optimized design for the purpose of comparison. Additionally, how are these constraints defined and what is their lower and upper limits? Please provide the details on how the fatigue damage calculation is done? What are the parameters, properties, and lower and upper limits defined as a design constraints? Similar to fatigue damage and frequencies, the details of other constraints, the way they are calculated and the upper and lower limits of each of them and how they are satisfied in the design are missing. Please provide these details.*

[**Authors**] Overall, the full and detailed comparison of the baseline and the optimized designs is not the primary scope of this work and would lead to an enormous document, probably hardly readable. We tried to overcome this by plotting and listing only the macro parameters of the optimization results, and make use as much as possible of our and others' existing publications. For example, readers interested in the details of rainflow counting, fatigue damage calculation procedure, eigenanalysis, etc. should refer to Bottasso et al., 2011, as indicated at the beginning of §2.2.

[revised manuscript text omitted]

---

## Author Comment (AC2) · 23 Mar 2016

**Combined Preliminary-Detailed Design of Wind Turbines**
**Detailed Replies to Reviewers**

**Reviewer #2**

We thank the reviewer for the detailed analysis and the fruitful comments. We also appreciated his recognition of the innovative content of the paper and the acknowledgement of the hard work required to develop the presented framework. The revised version of the manuscript is included at the end of this document, with a blue highlight to indicate all changes with respect to the previous version.

Below is a list of comments and answers to the **mandatory changes**:

1. [**Reviewer**] *Revise the statements that this is the first time optimization has included tower and rotor size to be more consistent with the current literature*

   [**Authors**] As also answered to Reviewer 1, the sentence has now been reformulated and a reference to WISDEM (Dykes et al., 2014) has been included.

2. [**Reviewer**] *I don't recall seeing a mention of any speed constraint on the rotor. Was there a tip speed constraint and how was it determined?*

   [**Authors**] The maximum allowable tip speed belongs to the list of given input data D, see Eq. 3, and it is held constant during the optimization process. The actual values are 72m/s for the 2.2MW and 90m/s for the 10MW, see Tables 1 and 6, respectively, and were selected by the industrial partner in the first case and by DTU in the second. The constraint operates in Eq. (2d) and it is active or not based on the design, i.e. rotor radius, solidity, rated power, wind turbine class etc.

3. [**Reviewer**] *Discuss the issue of rotor non-torque loads and how they were or were not included in the costing of the drive train components. It is assumed the drive train was designed for the steady loads, but it just needs to be made clear.*

   [**Authors**] Yes, in fact non-torque loads do not enter in the cost models of the drive train components. Following the reviewer's suggestion, extra text has now been added at the end of §2.3 to stress the importance of cost models.

4. [**Reviewer**] *In two places the term "under-designed" is used. It appears to mean under sized, but it is not clear. A more clear term should be used.*

   [**Authors**] The terms have been deleted and the text reformulated.

5. [**Reviewer**] *Plots such as Figure 6 showing blade design parameters should be labeled as "Blade span" instead of "Blade Length"*

   [**Authors**] Labels have been changed as suggested.

6. [**Reviewer**] *Plots such as Figure 5 showing tower parameters should have the independent variable of height on the x-axis and the dependent variables (such as thickness and diameter) on the y-axis to be consistent with the blade results.*

   [**Authors**] Plot axes have been reversed.

Below is a list of comments and answers to the **optional changes**:

1. *Reviewer] The practice of using software language statements (subroutine names) in the equations documenting the optimization process seems to be growing, especially in this area of system optimization. While it may provide better understanding to the practitioners of this area of expertise, it is not as clear to those (like this reviewer) who are outside the area. Consider revising the equations to use more traditional mathematical symbols and descriptors.*

   [**Authors**] We agree with the two reviewers that the description of system optimization algorithms is not simple. We chose to be coherent with our previous publications on wind turbine optimization (Bottasso et al, 2011, Bottasso et al, 2013, Bottasso et al., 2014a, Bottasso et al., 2015). Both in those publications and here, we always try to support the formal algorithmic description with text and figures, in order to balance between clarity and formality. We clearly welcome any specific suggestion to further improve our current descriptions of these complex algorithms.

2. *[Reviewer] On page 5, the first paragraph, there is a statement that no dependence of H is assumed for the 50 year storm wind speed. That sound ominous and could use more explanation.*

   [**Authors**] To our knowledge, international standards do not relate average and storm wind speed values to a specific hub height, but rather to the wind turbine class. In order to optimize hub height, we therefore had to move from an average wind speed prescribed by the class to a value linked to vertical shear. This resembles a site specific optimization framework. Regarding the storm wind speed, we decided to follow the class indication, and therefore we considered a constant value. This is clearly a designer's decision and a different choice, which would be trivial to implement, would have been possible and could have led to different solutions.

   Text was slightly expanded to better explain this point.

3. *[Reviewer] Better define the term "un-exceedable loads".*

   [**Authors**] Text has been adjusted to explain the possibility of freezing the load envelope for some of the components of the wind turbine. This may occur during studies where some parts, for example the hub or the nacelle, cannot be redesigned (for example, because the designer wants to re-use some existing components, or any other reason).

4. *[Reviewer] Discuss the option to introduce the LI designs through a distributed change in blade pitch (twist) rather than through a single offset.*

   [**Authors**] This suggestion for future work has now been added at the end of §3.2.2.

5. *[Reviewer] On page 7 it would be useful to describe each IEC DLC briefly rather than just referring to each number.*

   [**Authors**] The description came a few lines later. The order of the sentences is now changed to highlight the description of the various DLCs.

6. *[Reviewer] Also on p. 7, explain how close the "close to be active" the frequency constraints are.*

[**Authors**] The text included a small mistake, as a check of the data showed that the constraints were actually active. The text has now been changed accordingly.

7. *[**Reviewer**] The design driving load cases that provided active constraints are very nicely listed and discussed. Consider creating a table that summarizes them for each design size. These are very interesting results.*

[**Authors**] A publication from Croce et al. is under submission, where the detailed comparison of several design solutions for the 10 MW wind turbine is presented. Extensive load analyses will be part of that paper.

[revised manuscript text omitted]

---

## Author Response (AR1)

**Combined Preliminary-Detailed Design of Wind Turbines**
**Replies to Editor in Chief**

Thank you for your comments to our work. We have now modified the manuscript, as detailed in the following:

1. **[Reviewer]** *Several references are not easily accessible through a link or a DOI. Among those are: Bottasso C.L., Croce A.: Cp-Lambda: User's Manual, Dipartimento di Scienze e Tecnologie Aerospaziali, Politecnico di Milano, 2006–2016, but there are many more. Please find links to those (particularly manuals, conference papers, reports), such that the readers can easily access that material.*

   **[Authors]** We have eliminated the reference to Cp-Lambda's user's manual. We have provided doi for all papers, when available. We have refrained from providing links, as these are typically temporary and they might not be necessarily available in the future.

2. **[Reviewer]** *A direct link to Chaviaropoulos InnWind Report (which is 1.23, not 1.2.3) must be present.*

   **[Authors]** We have provided a link, as requested.

3. **[Reviewer]** *The associate editor expresses concern that the term "integrated optimization" is not technically completely correct. Add a sentence in the manuscript explaining the difference.*

   **[Authors]** We do not completely agree with the statement of the Associate Editor. Our procedure is not monolithic (which would mean solving for all design variables in one shot), but it is indeed integrated. Integrated means that the disciplines are able to mutually influence themselves: a change in the aerodynamics is reflected in a change in the structures, and viceversa (and similarly for the controls). This is explained at length in the paper, initially in the introduction, then in great detail in the technical sections, and finally the concept is repeated again in the conclusions. In addition, our procedure is not sequential. The term "sequential" implies cascading effects from a step to the next, but not the opposite. This does not happen in our procedures: although certain steps are done in sequence (because this allows for a much greater computational efficiency than a monolithic one-shot approach), there is always in our algorithms the feedback effect of the later steps on the earlier ones. We have paid great attention in the formulation of the text to make these concepts as clear as possible, and we believe that the reader will be able to correctly appreciate this aspect of the formulation. This is also well captured by one of the sentences of the conclusions, which states: "Although broken down in sequential steps, the overall iterative procedure results in an integrated algorithm, where changes in any one discipline (aerodynamics, structures, controls) influences the others." There are numerous other parts in the text that explain this concept in a way that we believe is clear and accurate.

4. **[Reviewer]** *The sizes of the labels of the axes, the numbers on the axes and the legend in figures 3 to 10 are not large enough. Please adjust.*

   **[Authors]** Size of the labels have now been changed.